# Optical-domain spectral super-resolution via a quantum-memory-based time-frequency processor

Mateusz Mazelanik[1,2 ✉], Adam Leszczyński[1,2] & Michał Parniak [1,3 ✉]

Existing super-resolution methods of optical imaging hold a solid place as an application in natural sciences, but many new developments allow for beating the diffraction limit in a more subtle way. One of the recently explored strategies to fully exploit information already present in the field is to perform a quantum-inspired tailored measurements. Here we exploit the full spectral information of the optical field in order to beat the Rayleigh limit in spectroscopy. We employ an optical quantum memory with spin-wave storage and an embedded processing capability to implement a time-inversion interferometer for input light, projecting the optical field in the symmetric-antisymmetric mode basis. Our tailored measurement achieves a resolution of 15 kHz and requires 20 times less photons than a corresponding Rayleigh-limited conventional method. We demonstrate the advantage of our technique over both conventional spectroscopy and heterodyne measurements, showing potential for application in distinguishing ultra-narrowband emitters, optical communication channels, or signals transduced from lower-frequency domains.

[1] Centre for Quantum Optical Technologies, Centre of New Technologies, University of Warsaw, Banacha 2c, 02-097 Warsaw, Poland. [2] Faculty of Physics, University of Warsaw, Pasteura 5, 02-093 Warsaw, Poland. [3] Niels Bohr Institute, Universtiy of Copenhagen, Blegdamsvej 17, 2100 Copenhagen, Denmark. ✉email: m.mazelanik@cent.uw.edu.pl; m.parniak@cent.uw.edu.pl

Optical spectroscopy is an indispensable tool in the studies of matter and light, including chemistry[1,2], astronomy[3], biology and medicine[4], metrology, and more. Yet, the resolution of all state-of-the-art methods, such as grating-based[5,6] and Fourier spectrometers[7–9], is subject to the Fourier limit. Methods of beating analogous Rayleigh limit are widely known in the context of imaging and include modifying or exploiting very specific properties of the source or illumination[10–13], which is often impossible to implement. Furthermore, even though the Rayleigh limit has been originally formulated in the context of a spectroscope[14], super-resolution methods of spectroscopy are scarce and limited to laser spectroscopy[15]. The task and challenges of fluorescence spectroscopy are starkly different: a typical emitter provides only a small photon number per single spectro-spatial mode, which is a strong incentive to seek quantum-enhanced protocols.

In imaging, the modern understanding of the Rayleigh limit is formulated as a vanishing information about separation distance between two sources for small separations, termed the "Rayleigh curse"[16]. Tsang et al.[16] have noticed that the quantum Cramér-Rao bound (Q-CRB), which identifies maximum information available in the optical field, is not saturated by traditional methods. This work inspired new methods that allow for reaching the quantum limit[17–23], most recently also in the time domain[24,25]. The idea to apply an analogous protocol for spectroscopy faces challenges, especially if operation with narrowband optical signal is desired. Yet, the inspiration can be drawn from another domain: in the nuclear-magnetic resonance (NMR) spectroscopy a quantum memory can lead to increased resolution when sensing with color centers in diamond[26–28]. A general framework for those experiments has been formulated by Gefen et al.[29] who identified specific quantum measurements that facilitate super-resolution. A resulting idea is therefore to use a quantum memory to achieve spectral super-resolution in the optical domain.

Here we bring the quantum-inspired super-resolution methods to the spectral domain and demonstrate a device that can resolve frequency differences of two emitters with precision below the Fourier limit. Here, by analogy to modern works on imaging, we understand this limit as a vanishing information about frequency separation between two spectral lines. Our method utilizes a Gradient Echo Memory (GEM) with built-in time-frequency processing capabilities which we program to realize a pulse-division time-axis-inversion (PuDTAI) interferometer. Our protocol operates in the ultra-narrowband domain, achieving a resolution of 15 kHz with simultaneous super-resolution enhancement factor of $\mathfrak{s} = 20 \pm 0.5$ which means that about 20 times less photons are needed to achieve the same resolution as direct spectroscopy under the same experimental conditions. Our work not only establishes a new super-resolution spectroscopy method, but also provides an high spectral resolution in absolute terms. Fundamentally, this super-resolving method exploits the spectral information already present in light, not requiring specific properties of source or illumination. Such performance is achieved by employing a spin-wave quantum memory to fully extract both phase and amplitude from the optical field. This is enabled by the long coherence time of the memory that allows us to capture, process, and release the light thus allowing optimal detection.

## Results

**Information in spectral resolution**. Let us first set the stage of the problem and introduce the theoretical framework to evaluate the super-resolution enhancement. Two mutually incoherent sources with equal brightness emit photons with the same spectral envelope $\tilde{\psi}(\omega)$ that is assumed to be known, and is the Fourier transform of the time-domain envelope $\psi(t)$. Such scenario is relevant both for the cases of fluorescence (with photon pulses being determined by spontaneous emission, assuming fast excitation and no inhomogenous broadening) and scattering (such as Raman scattering, where we may need to consider relative phases of ground-state coherences). It is also an optical analog of the radio-frequency-domain physics present in spectrum analyzers and nano-NMR[29]. A more conventional, but equally relevant scenario is when the linewidths of two sources are very narrow, and the resolution is limited by an aperature, which sets the transfer function $\tilde{\psi}(\omega)$.

The two sources in question have slightly different central frequencies $\omega_- = \omega_0 - \delta\omega/2$ and $\omega_+ = \omega_0 + \delta\omega/2$, where the frequency separation $\delta\omega = \sigma\varepsilon$ is smaller than the width $\sigma$ of the spectral envelope $\tilde{\psi}(\omega)$ that for the normalized separation parameter $\varepsilon$ translates to $\varepsilon \ll 1$. The spectroscopist's task is to estimate the separation $\varepsilon$ with maximum efficacy—to obtain maximum information about the separation per collected photon.

In the conventional approach, a spectrometer that measures the normalized spectral intensity $\tilde{I}(\omega) = \frac{1}{2}\left(|\tilde{\psi}(\omega - \delta\omega/2)|^2 + |\tilde{\psi}_-(\omega + \delta\omega/2)|^2\right)$ is used to collect data that is then processed to estimate $\delta\omega$, for example by fitting the theoretical curve. Similarly, a Fourier spectrometer measures the second-order autocorrelation function, which is then Fourier-transformed to yield $\tilde{I}(\omega)$. Notably, from the informational point of view, one may obtain precision of $\delta\omega$ much greater than $\sigma$, simply by collecting enough statistics. Therefore, to quantify the metrological advantage, we need to consider photon shot noise resulting from finite statistics, for which the typical asymptotic behavior is $\Delta^2\hat{\varepsilon} \propto 1/N$ with $N$ being the number of collected photons. This behavior is captured by the Cramér-Rao bound, which limits the maximum achievable precision for any locally unbiased estimator of the normalized separation $\hat{\varepsilon}$[30]:

$$\Delta^2\hat{\varepsilon} \geq \frac{1}{N\mathcal{F}_\varepsilon}, \quad \mathcal{F}_\varepsilon = \int \frac{1}{p_\varepsilon(x)}\left(\frac{\partial}{\partial\varepsilon}p_\varepsilon(x)\right)^2 dx, \quad (1)$$

where $\Delta^2\hat{\varepsilon}$ is the variance of the estimator, $\mathcal{F}_\varepsilon$ is the Fisher information, $N$ is the number of independent single photons used, and $p_\varepsilon(x)$ represents the measurement outcome distribution parametrized by the true separation value $\varepsilon$. For Gaussian envelopes

$$\tilde{\psi}_\mathcal{G}(\omega) = \left(\sqrt{2\pi}\sigma\right)^{-1/2}\exp\left(-\frac{\omega^2}{4\sigma^2}\right) \quad (2)$$

the Fisher information for DI in case of small separations $\varepsilon \ll 1$ can be approximated as $\mathcal{F}_{\mathrm{DI}} \approx \varepsilon^2/8$. This unfavorable scaling has been termed the Rayleigh curse, therefore we consider DI to be a Rayleigh-limited method. Physically, this reflects the fact that it is extremely hard to estimate the separation of two noisy, overlapping spectral lines as shown in the inset in Fig. 1. With more statistics, one may be able to do it, but the per-photon information $\mathcal{F}_{\mathrm{DI}}$ is small. At the same time the quantum Fisher information (QFI)[31] that yields a measurement-strategy-independent precision bound for a given estimation task turns out to have constant value $\mathcal{F}_\mathrm{Q} = 1/4$, independent of $\varepsilon$. Such analysis leads to the conclusion that the direct spectroscopy scenario, which is an analogy for real space direct imaging (DI) of two incoherent sources, is not optimal[16]. Hence, any strategy with Fisher information above $\mathcal{F}_{\mathrm{DI}}$ can be considered a super-resolving method beating the Rayleigh limit, or equivalently the Fourier limit in spectroscopy.

These observations have led to an extensive search for different measurement schemes that will approach the ultimate bound given by Q-CRB and will not suffer from the Rayleigh curse. Most

of those super-resolution measurement schemes are based on the following observation: a shifted Gaussian mode function $\tilde{\psi}_{\mathcal{G}}(\omega \pm \delta\omega/2)$ for $\varepsilon \ll 1$ can be approximated as an unshifted component $\tilde{\psi}_{\mathcal{G}}(\omega)$ with a small correction of the first Hermite-Gaussian function:

$$\tilde{\psi}_{\mathcal{G}}(\omega \pm \varepsilon\sigma/2) \approx \tilde{\psi}_{\mathcal{G}}(\omega) \pm \frac{\varepsilon}{4}\tilde{\psi}_{\mathcal{HG}}(\omega) \qquad (3)$$

$$\tilde{\psi}_{\mathcal{HG}}(\omega) = \frac{\omega}{\sigma}\left(\sqrt{2\pi}\sigma\right)^{-1/2}\exp\left(-\frac{\omega^2}{4\sigma^2}\right). \qquad (4)$$

From this, we see that only the antisymmetric component $\tilde{\psi}_{\mathcal{HG}}(\omega)$ carries the information about the separation, thus filtering out the (orthogonal) symmetric mode $\tilde{\psi}_{\mathcal{G}}(\omega)$ leads to massive improvement of signal-to-noise ratio and boosts up the separation estimation precision.

**Protocol**. The idea for achieving sub-Fourier performance in our spectrometer via the PuDTAI protocol relies on engineering a measurement that can split the signal pulse into symmetric and antisymmetric combinations with respect to the mean time or frequency. The protocol that realizes this projective measurement is inspired by a technique known from conventional (real space)

super-resolved imaging called SLIVER (superlocalization via image-inversion interferometry)[21,22,32–34], where an image is inverted in one arm of a Mach–Zhender interferometer. The Fisher information associated with photon detection probabilities in the symmetric $\mathcal{P}_{\mathcal{G}}$ and antisymmetric $\mathcal{P}_{\mathcal{HG}}$ ports exhibits extraordinary sensitivity for small ($\varepsilon \ll 1$) separation estimation $\mathcal{F}_{\text{SLIVER}} \approx \frac{1}{4} - \frac{\varepsilon^2}{32}$ and for $\varepsilon \to 0$ approaches $\mathcal{F}_{\text{Q}}$.

In our protocol, schematically depicted in Fig. 1, we divide the signal in half, rather than splitting the signal to prepare two identical copies. This method while being slightly different from SLIVER, achieves the same sensitivity to source separation and inherently, as an additional feature, allows use with $N$-photon states[35]. This framework assumes prior knowledge of the mean frequency, also known as the source centroid. This assumption is practically valid, as an adaptive strategy can be used where the centroid is first estimated with desired precision using direct detection[36].

**Temporal imaging in GEM**. Before introducing the details of the PuDTAI protocol, let us first discuss a simpler case of DI spectrometer based on an atomic Gradient Echo Memory (GEM)[37–39] that is realized in the same GEM setup schematically depicted in Fig. 2a. In this setup, a signal pulse with a slowly varying amplitude $\mathcal{A}(t)$ enters the atomic cloud placed in the

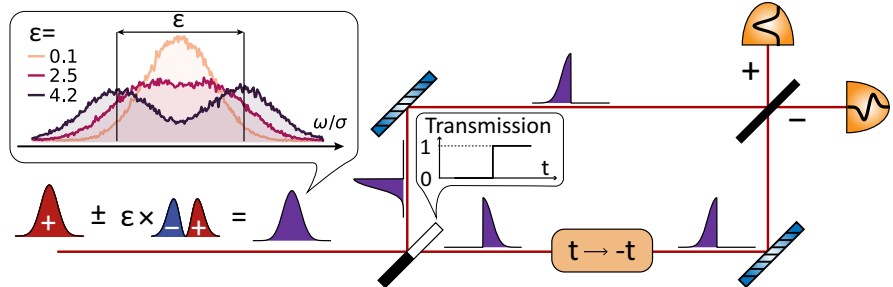

**Fig. 1 The idea for time-inverting interferometer for super-resolving spectroscopy.** The task is to measure a separation between two spectral lines. Separation is smaller than the linewidths, which are determined by the properties of the source itself or by the temporal aperture of the spectroscopic measurement device. As shown in the inset, photon shot noise (stochastic counts) prevent an accurate estimate of separation when two lines overlap strongly. For small separations, however, the information about the separation is all contained in the antisymmetric part of the signal pulse. To extract this information, the signal pulse is sent to the pulse-division time-axis-inversion (PuDTAI) interferometer that performs decomposition onto symmetric and antisymmetric components. First, a time-dependent transmission mirror splits the pulse in half between the two interferometer arms. One of the arms includes a time-inversion device that mirrors the first half of the pulse. Finally, the two components are combined on a beamsplitter, and the resulting symmetric (+) and antisymmetric (−) projections are detected.

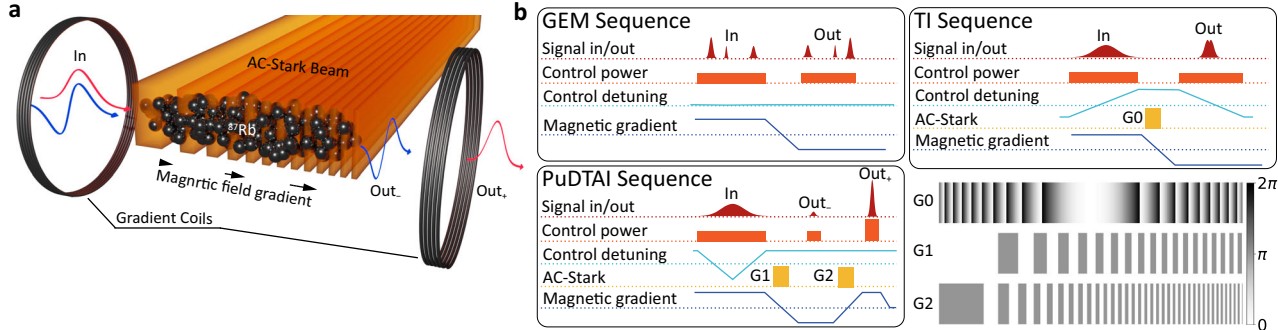

**Fig. 2 Experimental setup and sequence. a** The Gradient Echo Memory (GEM) device enhanced with the spatial AC-Stark modulation, used to store, process, and release the signal pulses. The same setup is used to implement the simple GEM, the temporal imaging (TI), and the PuDTAI protocol, for which the pulses are shown here. In the PuDTAI interferometer implementation, a symmetrically-chirped control field maps the signal pulse onto atomic ground-state coherence of $^{87}$Rb atoms and simultaneously implements the time-dependent-transmission mirror and time-inversion transformation. AC-Stark phase modulation with chirped square-wave gratings (AC-S Gratings) is used to implement the beamsplitter operation and sequentially read-out the resulting interferometer ports (Out$_-$, Out$_+$). **b** Experimental sequences and AC-Stark beam patterns (G0–G2) used to implement the basic (pass through) GEM protocol, TI spectrometer and the PuDTAI interferometer. The full sequence for PuDTAI protocol and relevant atomic levels are shown in Fig. 7 in the "Methods" section.

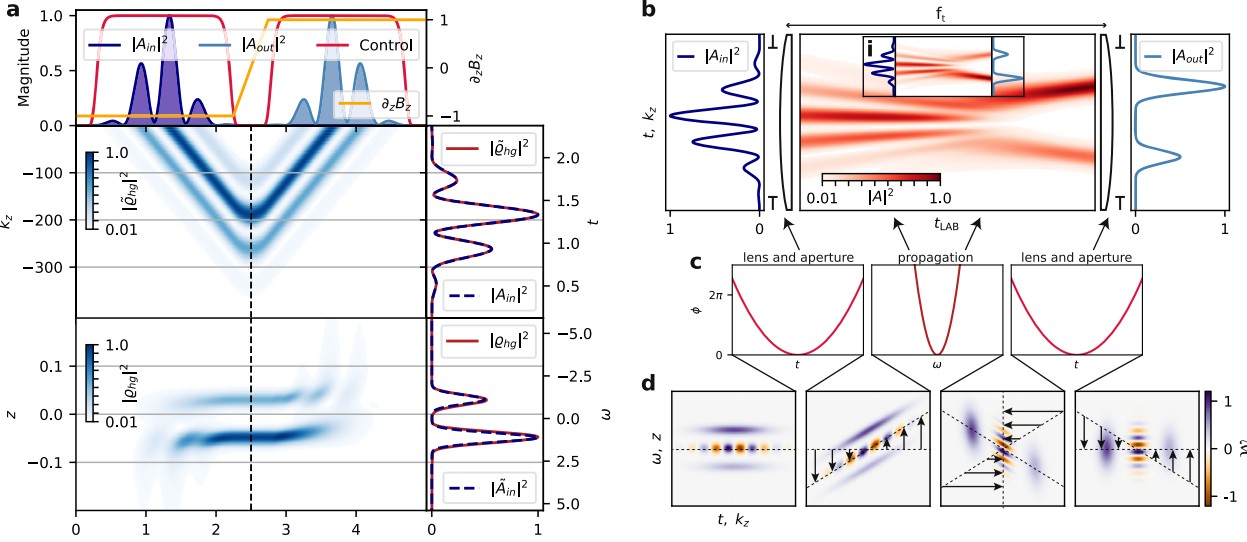

**Fig. 3 Temporal imaging in GEM. a** Simulation of GEM protocol for signal ($\mathcal{A}_{in}$) with two spectral components with finite bandwidth. The maps show the evolution of the atomic coherence ($\varrho_{hg}$) in the wavevector (top) and real space (bottom). The frequency-to-space and time-to-wavevector mapping feature of GEM protocol is visible on the cross-sections (taken along the dashed lines) of the coherence maps that are compared to the temporal and spectral shape of the stored optical field. **b** Temporal far-field imaging using two time-lenses separated by temporal propagation. The output signal is a Fourier transform of the input signal. The temporal aperture clips the input signal and sets the frequency resolution limit. The inset **i** shows propagation of the same input signal through a similar TI setup but with a lens with negative focal length followed by backward temporal propagation, resulting in the inverted spectrum at the output. **c** Phase shapes that are sequentially imprinted on the input signal to implement the lens-propagation-lens TI setup. **d** Time-frequency and space-wavevector Wigner representation ($\mathcal{W}$) of the input signal at each stage of the TI setup. The arrows represent the transformations of the phase-space upon each transformation (lens or propagation).

magnetic field gradient along the propagation axis $g = \partial_z B_z$. At the same time, the cloud is illuminated by a strong, co-propagating control field $\mathcal{C}$ that due to two-photon interaction maps the signal field onto atomic coherence described by the off diagonal terms $\varrho_{hg}(z)$ of the spatially dependent $\{|g\rangle, |h\rangle\}$ two-level atom density matrix $\hat{\varrho}(z)$. To restore the signal light, the gradient is switched to the opposite $g \rightarrow -g$, and the control field pulse is applied again, as shown in the corresponding sequence in Fig. 2b. In Fig. 3a, we show a numerical simulation of the GEM protocol for a signal pulses with two spectral components of different amplitudes. The density maps below the time trace show the evolution of the atomic coherence in the real space and wavevector coordinates. The presence of the magnetic field gradient enables the crucial feature of the GEM protocol which is the spectro-spatial mapping that links spectral components of the signal pulse with the amplitude of the atomic coherence along the ensemble $\tilde{\mathcal{A}}(\omega) \leftrightarrow \varrho_{hg}(z)$. At the same time, in the time domain, the pulse shape is transferred to wavevector-space components of the coherence $\mathcal{A}(t) \leftrightarrow \tilde{\varrho}_{hg}(k_z)$, where $\tilde{\varrho}_{hg}(k_z)$ is the Fourier transform of the real space coherence. This correspondence is directly visible on the cross-sections of the atomic coherence maps (right panels) taken along the dashed lines where we also show the optical field magnitudes in the time and spectral domain.

The DI imaging spectrometer utilizes a temporal imaging (TI) technique to perform a Fourier transform of the signal pulse and thus map a spectrum of the signal pulse onto the temporal shape, as illustrated in Fig. 3b. This is achieved by applying sequentially temporal and spectral phase modulations corresponding to lens-propagation-lens components. A time lens corresponds to applying a quadratic phase in time, while propagation is achieved by imprinting a quadratic phase in spectrum onto the signal pulse as shown in Fig. 3c[40]. In GEM, the temporal phase modulation is achieved by chirping the control field that controls

the light-coherence mapping process. The temporal phase modulation is this way reflected in the phase of the coherence in the wavevector space ($k_z$). On the other hand, the phase modulation in the spectral domain is achieved by Spatial Spin-wave Modulation (SSM)[41–43], that utilizes spatially varying ac-Stark shifts to imprint arbitrary phase profiles onto the coherence in the real space coordinate ($z$). The ac-Stark shifts are caused by an intensity-shaped beam that illuminates the atoms from the side (see Fig. 2 for the geometry and the actual sequence of the experiment) during the storage time.

The most convenient way to understand the evolution of the optical field or the corresponding atomic coherence at each step of the protocol is to investigate a phase-space quasi-probability given by the Wigner function

$$\mathcal{W}_Q(q, p) = \frac{1}{\sqrt{2\pi}} \int Q(q + \xi/2) Q^*(q - \xi/2) \exp(-ip\xi) \mathrm{d}\xi \quad (5)$$

where the quantity of interest $Q$ and the conjugate variables $\{q, p\}$ are either $\mathcal{A}$, $\{t, \omega\}$ for which it becomes a chronocyclic Wigner function[44] describing the signal pulse or $\varrho_{hg}$, $\{k_z, z\}$ for the atomic coherence.

In Fig. 3d, we show evolution of the chronocyclic Wigner function for the input signal propagating through the TI setup from Fig. 3b. A temporal phase modulations $\mathcal{A}(t) \rightarrow \mathcal{A}(t) \exp(i\phi(t))$ reshapes the chronocyclic Wigner function $\mathcal{W}_\mathcal{A}(t, \omega)$ in the $\omega$ direction, at the same time the corresponding coherence quasi-probability is reshaped in the $z$-direction $\mathcal{W}(k_z, z) \xrightarrow{\phi(t)} \mathcal{W}(k_z, z')$. For example, in Fig. 3d, the quadratic phase profile that implements the time-lens stretches the $\omega$ axis on the $\mathcal{W}_\mathcal{A}(t, \omega)$ map linearly with $t$ as marked by the arrows.

The spectro-spatial mapping enables implementation of spectral phase operations by phase-modulating the coherence in real-space coordinates $\varrho_{hg}(z) \rightarrow \varrho_{hg}(z) \exp(i\chi(z))$. These are

linked to $k_z$-direction reshaping of the quasi-probability $\mathcal{W}(k_z, z) \xrightarrow{\chi(z)} \mathcal{W}(k_z', z)$ and $t$-axis transformation of the chrono-cyclic Wigner function. This is visible in the third panel of Fig. 3d corresponding to propagation transformation realized by a quadratic spectral phase profile. In the last panel, the second time-lens operation completes the 90° rotation of the phase—the Fourier transform of the input signal amplitude $\mathcal{A}(t)$ is complete. In practice, however, one may omit the last step as it only corrects the phase of the resulting output signal that is (the phase) irrelevant in the case of phase insensitive detection as photon counting.

The discussed TI setup not only serves as an introductory example but is implemented experimentally in our memory[37] to serve as DI reference to our super-resolution protocol, which we describe below.

**PuDTAI interferometer**. Finally, within the introduced framework, we are able to realize both the time-dependent-transmission mirror that divides the pulse and performs a time-inversion as well as the beamsplitter that combines the two components and performs the final projection. The sequence of operations performed using the quantum memory is shown in Fig. 2b.

The step-by-step evolution of the quasi-probability $\mathcal{W}(z, k_z)$ in the PuDTAI protocol for two input pulse shapes $\psi_{\mathcal{G}}$ and $\psi_{\mathcal{HG}}$ is presented in Fig. 4b. First, similarly to TI, a temporal phase modulation with $\phi(t) = -\alpha t|t|/2$, corresponding to $z \to z' = z + \alpha|k_z|$ transformation in the phase-space implements the time-dependent-transmission beamsplitter and simultaneously performs the time-inversion of the first part. In fact, such modulation represents a dual time-lens that is half convex, half concave i.e., the focal length of the first half ($t < 0$) is positive when for the second part ($t > 0$) is negative. The Wigner function maps (dark background) illustrate that the two parts of the pulse are mapped symmetrically on the opposite sides of the $k_z$ space. For practical reasons, the temporal phase modulation is accompanied with Cassegrain-type temporal aperture to prevent mapping of the central part of the pulse (around $t = 0$), the aperture is represented by a black dashed line on the signal pulses time traces (first row of Fig. 4b). The (minor) influence of the aperture on the protocol's performance is discussed in the Methods section.

Next, by performing $k_z$-direction splitting transformation in the phase-space we overlap the two parts and make them interfere. This is achieved by imprinting a $\pi$-depth square-wave grating (G1) with linearly increasing grating wavevector $k_g = \kappa z$ (Fig. 2b). The chirped grating transfers the spin-wave into diffraction orders with $z$-dependent wavevector spacing $k_z \to k_z' = k_z \pm \kappa z$. Let us now focus on the relevant diffraction orders: the minus-first in the negative time part and the first one in the positive time part. Those two orders in the language of temporal imaging are corresponding to forward and backward propagation respectively. If we combine this with the dual-lens operation implemented in the first step we achieve dual-far-field temporal imaging setup that performs forward Fourier transform of the first half of the pulse and backward Fourier transform of the second half. Moreover, as the resulting parts overlap in the Fourier domain they interfere as represented in Fig. 4a. For this to happen, the propagation distance must be equal to the focal length of the time lens, the condition is met for the grating chirp parameter $\kappa$ equal to the inverse of the storage chirp $\kappa = 1/\alpha$. The third row of Fig. 4b represents the Wigner function after the application of the grating. Here we see that the two signal pulse components are forced by the grating (G1) to interfere at $k_z = 0$ coordinate (the symmetry axis of the quasiprobability distribution). By taking a $z$-direction integral of the quasi-probability around $k_z = 0$ we retrieve light intensity at the output of the

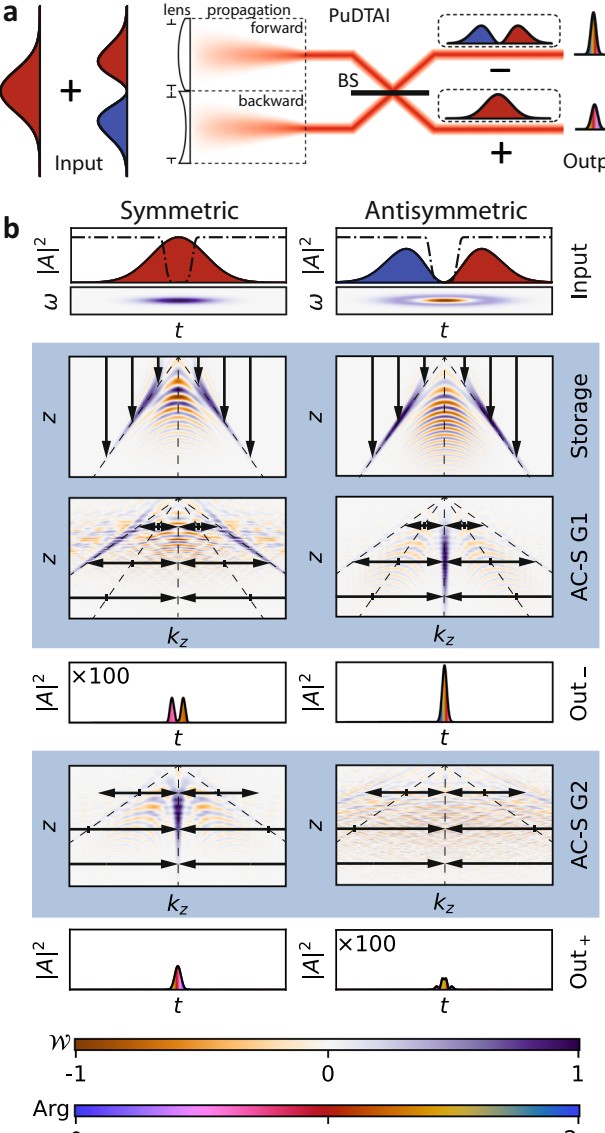

**Fig. 4 Step-by-step realization of the protocol for symmetric and antisymmetric input signals.** a Schematic representation of the PuDTAI interferometer superimposing the divided signal in the Fourier space. The color fill represents the phase (Arg) of the signal field envelope, BS beamsplitter. **b** The corresponding spin-wave evolution in $(z, k_z)$ space is represented by Wigner function maps ($\mathcal{W}$). The time-frequency-domain Wigner maps are shown for reference. Thanks to the symmetrically-chirped control field used in GEM protocol (Storage) the two halves of the signal pulse are mapped in a symmetric manner on separate Fourier ($k_z$) components while overlapping in the longitudinal coordinate $z$. The black dashed lines represent the temporal aperture that removes the central part of the singnal pulse during the mapping. The spatially-resolved AC-Stark phase modulation (AC-S) with chirped square grating splits the spin-waves in the $k_z$ direction and makes the two halves interfere at central $k_z$ coordinate —the antisymmetric port of the interferometer which is then read-out by control field pulse (Out_). The second (symmetric) port is restored by modulating the spin waves again with a second chirped square-wave grating with opposite phase (AC-S). The symmetric port is finally retrieved (Out_+).

memory. Moreover, the phase between the two interfering components can be adjusted by shifting the grating phase by $\zeta$: $\mathrm{sq}(\kappa z^2) \to \mathrm{sq}(\kappa z^2 + \zeta)$. For $\zeta = 0$ the interference for symmetric input shapes as $\mathcal{G}$ is destructive, while for antisymmetric $\mathcal{HG}$ it becomes constructive, as can be seen in Fig. 4a. To access the

**Table 1 Summary of the temporal and spectral phase modulations used to implement both PuDTAI and DI spectrometer.**

| Function | Type | Formula | Wigner space |
|---|---|---|---|
| Lens | $t \leftrightarrow k_z$ | $\phi(t) = \frac{\alpha}{2} t^2$ | $z \rightarrow z - \alpha k_z$ |
| Propagation | $z$ | $\chi(z) = \frac{\beta}{2} z^2$ | $k_z \rightarrow k_z + \beta z$ |
| Dual lens | $t \leftrightarrow k_z$ | $\phi(t) = -\frac{\alpha}{2} t\lvert t \rvert$ | $z \rightarrow z + \alpha\lvert k_z \rvert$ |
| Bidirectional propagation | $z$ | $\chi(z) = \mathrm{sq}(\kappa z^2)$ | $k_z \rightarrow k_z \pm \kappa z$ |

The $t \leftrightarrow k_z$ type of transformation means that it is implemented during the light-to-atoms mapping stage by temporal phase modulation of the control field, while the $z$ stands for the real-space phase modulation of the atomic coherence. The formula for the square wave pattern is $\mathrm{sq}(\xi) = \pi((-1)^{\lfloor \xi/\pi \rfloor} + 1)/2$. Each modulation has its corresponding Wigner space transformation that can be expressed in the $\{k_z, z\}$ or $\{t, \omega\}$ coordinates.

symmetric port the coherence is modulated again with similar square-wave grating (G2), but with doubled chirp parameter $k_g = 2\kappa z = \frac{2}{\alpha}z$, and shifted in phase by $\pi/2$. With this modulation, we simply reverse the previous temporal propagation step, and make it again but with the opposite phase between the two components. This is illustrated in the fifth row of Fig. 4, where we observe constructive interference for the symmetric input shape, and destructive for antisymmetric one. Finally, the area around $k_z = 0$ is remapped to light which constitutes the output from the symmetric port of the interferometer (last row of Fig. 4). Notably, the Wigner space domain transformations are the same for both input shapes, and the different results at the output are caused solely by the difference in the input shape. In Table 1 we provide a survey of the phase operations needed to implement the PuDTAI protocol. To summarize, the protocol realizes a $\pi/2$ $(-\pi/2)$ rotation of the Wigner function of the first (second) half of the signal pulse, and interferes with the two parts in the Fourier domain. Additionally, as the output have the Fourier domain mapped to the time axis, a longer signal pulses give shorter pulses at the output of the interferometer that leads to a higher signal-to-noise ratio when using noisy detectors. This provides an additional practical advantage while dealing with very narrowband states of light.

**Experimental results**. To benchmark our protocol we artificially prepare signal pulses composed of two mutually incoherent spectral components

$$\tilde{\mathcal{S}}_\varphi(\omega) = \frac{1}{\sqrt{2}} \left( \tilde{\psi}_{\mathcal{G}}(\omega - \sigma\varepsilon/2) + e^{i\varphi}\tilde{\psi}_{\mathcal{G}}(\omega + \sigma\varepsilon/2) \right) \qquad (6)$$

where the random phase $\varphi \in [0, 2\pi[$ is drawn from uniform distribution. The corresponding chronocyclic Wigner function[44] of such signal distribution reads:

$$\mathcal{W}_S(t, \omega) = \psi_{\mathcal{G}}^2(t)(\tilde{\psi}_{\mathcal{G}}^2(\omega - \sigma\varepsilon/2) + \tilde{\psi}_{\mathcal{G}}^2(\omega + \sigma\varepsilon/2))$$

We send the pulses to the memory where they are processed by our pulse-division time-inversion interferometer. To read-out the contributions in symmetric and antisymmetric ports we use two sequentially applied pulses of the control field that are interleaved with the second AC-Stark modulation. The processed signal light coming from the memory is detected using single-photon counting module (SPCM) and photo-counts are time-tagged to identify the output ports (see Fig. 9a in the "Methods" section for the time-bin histogram). The mean single-shot signal contribution summed over the symmetric and antisymmetric port was set to be around $\bar{n} \approx 0.69$. After many experimental repetitions, we count the total contributions $N_-$, $N_+$ in the antisymmetric and symmetric port respectively and calibrate the maximum-likelihood estimator $\hat{\varepsilon}(N_-/N_+)$ based on theoretically expected counts ratio $N_-/N_+$.

The estimator is then used to estimate the value of the separation $\varepsilon$. For this, for each $\varepsilon$ value, we collected approx. $1.5 \times 10^5$ counts. To estimate the estimator variance we follow the bootstrapping technique: for each $\varepsilon$ setting, we randomly prepare $10^3$ sets of samples, each containing $1.5 \times 10^5$ total counts. The estimator is then evaluated on each set and the mean and variance of $\hat{\varepsilon}$ is calculated. In Fig. 5a, we show raw estimation results for the PuDTAI protocol and DI approach on a common $\langle \hat{\varepsilon} \rangle$ plot. The filled regions correspond to the estimation uncertainty given by the square root of the estimator variance and normalized to 10 processed photons. Both methods give similar values of the separation parameter, but as we expect the PuDTAI protocol vastly outperforms the DI over the whole $\varepsilon < 1$ range. The improvement expressed in variance ratio for the same experimental conditions is most prominent for small separation values and reaches about 20 for $\varepsilon = 0.08$. In other words, using the PuDTAI protocol we need, 20 times less photons to achieve the same precision.

The measurements in DI scheme are obtained by using ultra-narrowband far-field temporal imaging technique which we call quantum-memory temporal imaging (QMTI)[37]. The protocol is implemented in the same GEM device following the recipe from the introductory section. It performs a 90° rotation of the signal pulse Wigner space which results in spectrum-to-time mapping $\omega \rightarrow 2\alpha_{DI}t$, where $\alpha_{DI}$ is the control field chirp parameter corresponding to the time-lens magnitude. The output signal is detected using the same SPCM as in the case of PuDTAI, where counts corresponding to distinct spectral components have different arrival time (timetags). The counts distributions with already calibrated frequency labels are presented in Fig. 5d. From those, we estimate the separation using maximum-likelihood separation estimator $\hat{\varepsilon}_{DI}$. The variance of this estimator is calculated by following the same bootstrapping technique as in the case of the PuDTAI protocol. Figure 5b portrays the estimators' biases $\langle \hat{\varepsilon} \rangle - \varepsilon$ proving unbiasedness of both estimators even for small $\varepsilon$ and validating the results for variances.

In Fig. 5c, we compare the achieved precision with CRB for both schemes. Here we also show the ultimate bound set by the QFI ($\mathcal{F}_Q$) as well as bounds for ideal SLIVER protocol ($\mathcal{F}_{SLIVER}$) and idealized DI ($\mathcal{F}_{DI}$). The real bounds are placed by FIs that take into account experimental imperfection such as detection noise and finite bandwidth of the ensemble—$\mathcal{F}_{QMTI}$, as well as finite interferometer visibility and interferometer losses in the case of PuDTAI protocol—$\mathcal{F}_{PuDTAI}$ (see "Methods" section). We see that the PuDTAI protocol vastly outperforms the QMTI for normalized separations $\varepsilon < 1$ with maximum improvement in terms of variance of about 30 for $\varepsilon = 0.4$. Finally, we show the data used for estimation, to better illustrate the origin of sensitivity enhancement. In Fig. 5d the results of direct imaging are presented, and we observe that for $\varepsilon < 1$ the shape of the obtained spectrum remains close to a Gaussian. On the other hand, in Fig. 5e, we show total contributions to the antisymmetric ($N_-$) and symmetric ($N_+$) ports, as well as the ratio $N_-/N_+$ along with fitted model that is used to estimate the $\varepsilon$ without a direct need to know total source brightness.

## Discussion

We have demonstrated frequency-separation estimation which outperforms direct spectroscopy. The protocol operates in a previously unexplored regime of very narrowband light, which merits further discussion and comparison with other possible schemes. In Fig. 6, we compare several different approaches to measure the frequency difference of two sources. We characterize the gain achieved with our method via the super-resolution parameter $\mathrm{s}$ that is interpreted as a reduction of resources (number of photons) required to achieve the same resolution as

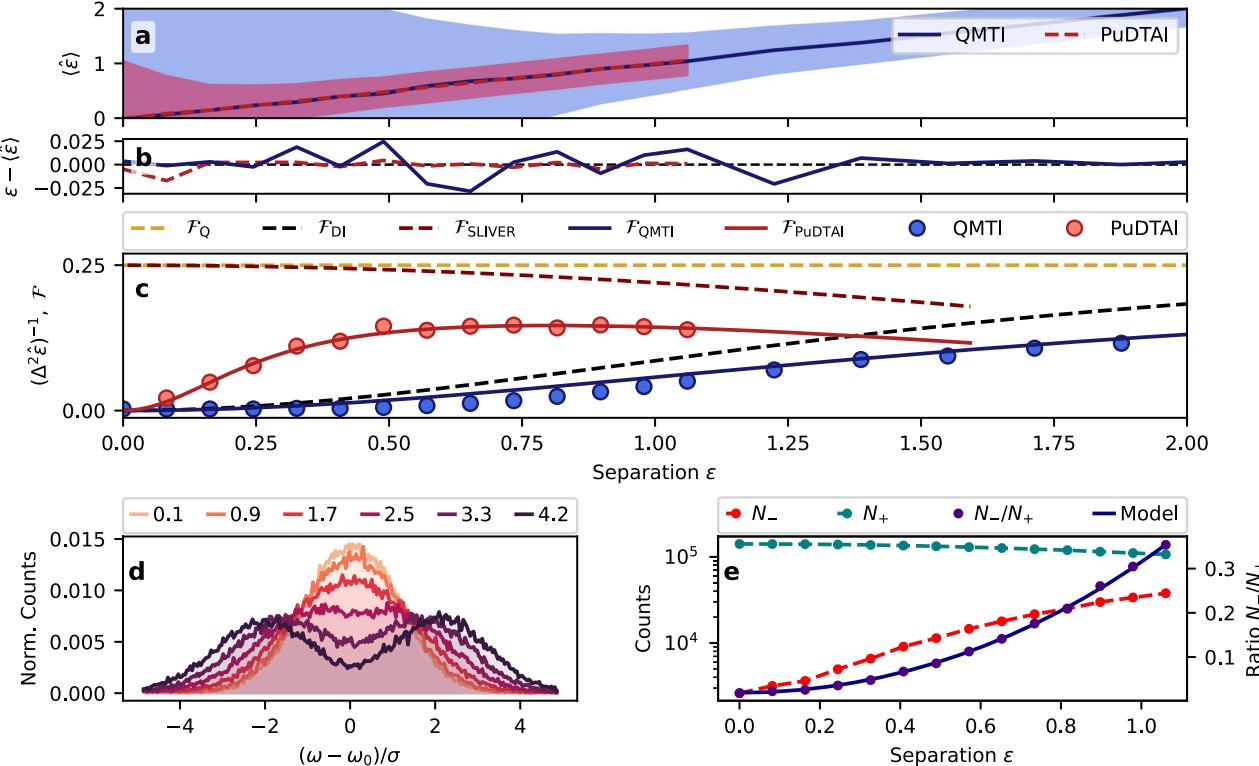

**Fig. 5 Separation estimation using QMTI and PuDTAI protocols. a** Raw $\varepsilon$ estimates obtained from DI approach (QMTI) and PuDTAI protocol. The shaded regions represent estimation uncertainty given by the square root of the estimator variance (SEM) normalized for 10 processed photons. **b** Estimator's biases for both schemes. **c** Estimation precision compared with idealized and real world CRBs given by corresponding Fisher information ($\mathcal{F}_Q$ Quantum Fisher information, $\mathcal{F}_{SLIVER}$ ideal SLIVER protocol, $\mathcal{F}_{DI}$ ideal DI spectrometer, $\mathcal{F}_{QMTI}$ Quantum memory temporal imaging, $\mathcal{F}_{PuDTAI}$ PuDTAI protocol). **d** Frequency-labeled single photon counts distributions obtained with QMTI spectrometer for different true separation values $\varepsilon$. **e** Contributions to the antisymmetric ($N_-$) and symmetric ($N_+$) port of the PuDTAI device and the ratio $N_-/N_+$ along with the calibrated theoretical model.

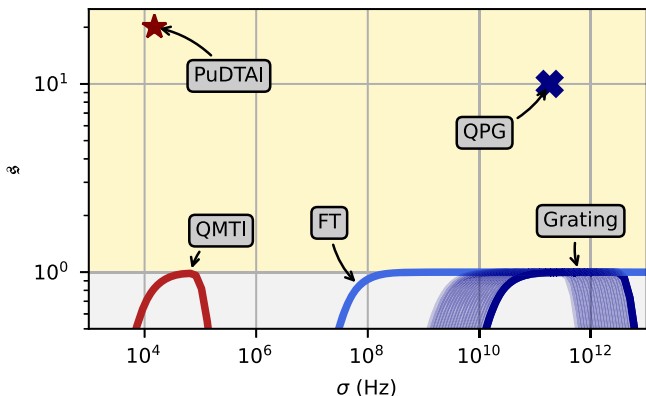

**Fig. 6 Comparison of super-resolution enhancement factor $\mathfrak{s}$ of different spectrometers and super-resolution spectroscopy techniques.** All conventional methods (Grating, FTS Fourier transform spectrometer) and the temporal-imaging method using our quantum memory (QMTI quantum-memory temporal imaging) are analogous to the DI approach and thus fall under the Fourier limit. Only two (QPG Quantum Pulse Gate[24] and PuDTAI—this work) beat the limit of $\mathfrak{s} = 1$ and thus achieve super-resolution. The curves represent exemplary realizations of a given spectrometer for the constant time-bandwidth product.

direct imaging spectroscopy with the same aperture $\sigma$. The $\mathfrak{s}$ for a given measurement scheme characterized by the Fisher information $\mathcal{F}$ is calculated as

$$\mathfrak{s} = \lim_{\varepsilon \to 0}(\mathcal{F}/\mathcal{F}_{DI}). \qquad (7)$$

While for the perfect case with $\mathcal{F} = \mathcal{F}_Q$ the super-resolution enhancement would reach infinity, this is practically never the case. In particular, it has been recently shown that effects such as finite visibility, noise, or cross-talk will always restore the $\varepsilon^2$ scaling[45], making the above limit well-defined. For instance, in the case of SLIVER with finite visibility we will have:

$$\mathfrak{s}_{SLIVER} = \frac{\mathcal{V}^2}{2(1 - \mathcal{V}^2)}. \qquad (8)$$

Thus, for a high visibility the enhancement may be immense, yet remains well-defined for small $\varepsilon$. The PuDTAI protocol follows a similar scaling, with the full formula given by Eq. (24) in the Methods section that yields $\mathfrak{s} = 20 \pm 0.5$.

All the conventional methods that include: grating and Fourier transform (FT) spectrometers along with far-field temporal imaging or inherently lossy scanning methods that employ cavity or EIT media fall into DI description and are Fourier-limited ($\mathfrak{s} = 1$). Surprisingly, there is only one more demonstration (Quantum Pulse Gate—QPG[24]) which yielded improvement over the conventional DI approach. Until now, the QPG approach, which in principle enables temporal-mode demultiplexing, allowed projecting only a single mode at a time. In our work, we are able to observe two modes (ports) in the same experiment. An important future challenge in both techniques is to allow truly multimode sorting.

While using DI (a spectrometer) for high bandwidth signals seems an obvious choice, one may conclude that for narrowband signals a balanced heterodyne (or homodyne) detection would perform better. The first obvious challenge is that one then needs a stable and narrowband local oscillator, but as our method uses a

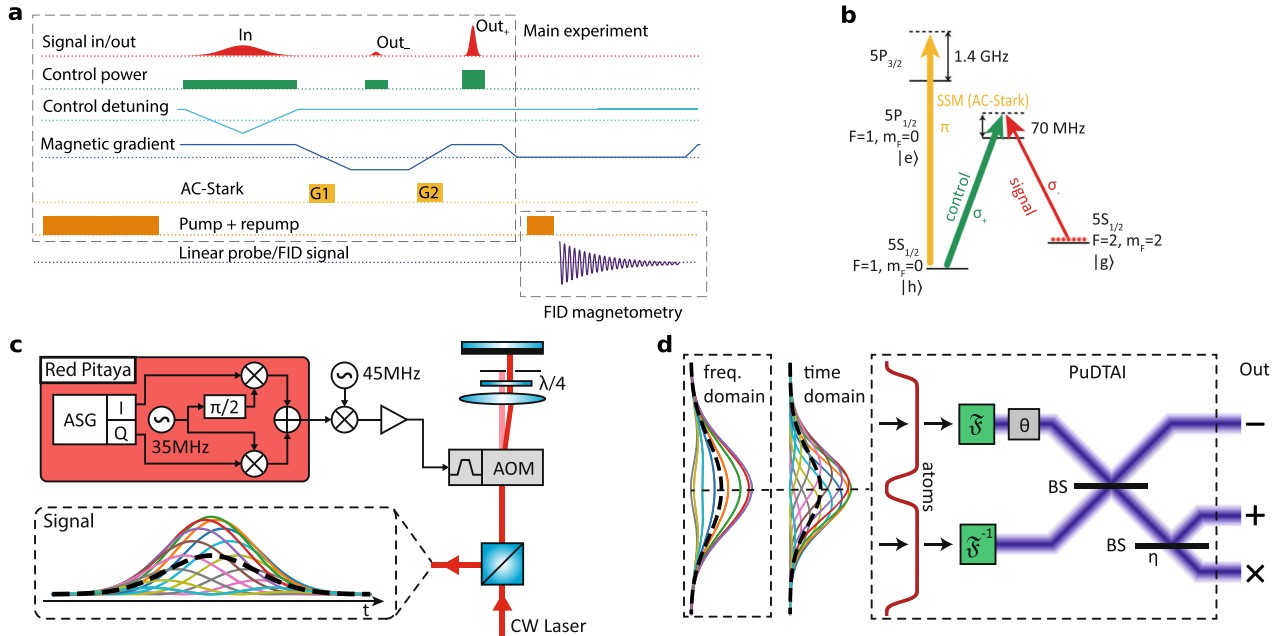

**Fig. 7 Experimental details and schematic representation of PuDTAI model. a** The sequence is separated into the main experiment part and the auxiliary magnetic field measurement part via free-induction decay (FID) measurement. In the main sequence, we show the signal input/output, the control field intensity and detuning, the GEM magnetic field gradient, the AC-Stark patterns with two grating patterns G1 and G2, and the pump/repump lasers that prepare the initial atomic state. In the FID magnetometry sequence, we show the same preparation as in the main experiment as well as the FID signal registered on a polarization-rotation detector. The signal is used to correct the mean magnetic field asynchronously. **b** The essential levels of $^{87}$Rb atoms used in the protocol as well as control, signal, and AC-Stark light fields with respect to atomic-level structure. **c** Experimental setup used to generate frequency-domain double-Gaussian signal $\tilde{\mathcal{S}}_\varphi(\omega)$. The in-phase (I) and quadrature (Q) signal components generated using the arbitrary signal generator module (ASG) are mixed with intermediate carrier frequency at 35 MHz using digital IQ-mixer. The resulting radio-frequency (RF) signal is then up-converted by 45 MHz with help of analog mixer. Finally, the RF signal with carrier frequency at 80 MHz is feed into acousto-optic modulator (AOM) in double-pass configuration that carves the light pulses (Signal) from continuous wave (CW) laser. **d** The input signal passes through a temporal aperture (atoms) and is split into two halves. The upper and lower parts are Fourier-transformed ($\tilde{\mathfrak{F}}$) (inverse transform—$\tilde{\mathfrak{F}}^{-1}$—for the lower part) and combined on the beamsplitter (BS) with relative phase set to $\theta$. The first output port (Out$_-$) is directly detected while the second port (Out$_+$) before detection experiences additional loss due to lossy ($\eta$) implementation of the beamsplitter.

control field, this requirement is effectively shared. However, when the sources brightness is low (below one photon per mode), the heterodyne detection scheme suffers strongly from shot noise. For the particular task of separation estimation, the shot noise will affect the estimation precision and makes it useless for measuring single-photon-level sources. Nevertheless, heterodyne (or homodyne) can overcome the Fourier limit for higher signal photon levels and for small separations as has been recently analyzed in detail for the spatial domain[46] ($\mathfrak{s} > 1$ for average 7.2 photons per mode and more), with the use of optimized data analysis, as opposed to simply looking at power spectral densities. Therefore, a purely photon-counting method is in general needed in the low light-level regime.

It is also worth considering the influence of fluctuations on our method. If the amplitude or phase of the signal varies over the single shot (temporal aperture), clearly the two parts of the pulse will not be able to interfere perfectly. Such a situation represents and unavoidable loss of information, and may be quantified via the reduced visibility $\mathcal{V}$ of interference. The result is therefore the same as already considered in the case of finite visibility caused by the device itself (see Eq. (8) and "Methods" section), yet the information here is already reduced in the signal itself.

The conjunction of the operation in the optical domain and the kHz-level resolution has promising application in the schemes that combine microwave or radio-frequency and optical domains[47]. Our method of spectroscopy can provide an optimal detection scheme for light that is transduced from the RF or microwave domain, without the sensitivity loss due to shot noise,

which is inherently present in the traditional heterodyne optical receivers.

We envisage that other platforms may be used to implement the protocol we proposed, also in different frequency regimes. For example, rare-earth-doped crystals such as Eu:Y$_2$SiO$_5$ offer narrow homogeneous absorption linewidth that can be dynamically broadened with an electric field[48] for the purpose of time-frequency-domain multimode storage[49]. Processing with AC-Stark shifts has also been demonstrated[50], and advances in embedding ions in waveguides promise good efficiency and noise properties.

Another approach has been very recently discussed theoretically by Shah and Fan[51], who considered a transformation device based on pulse shapers and electro-optic modulators. This approach may be very applicable where relevant devices can operate, which is in particular GHz-bandwidth regime in the telecom band of light. In this regime, no quantum memory is required, as the timescales involved are much shorter. Several other quantum-information protocols have been realized with similar means, also for quantum light, and the scheme is generally termed quantum-frequency processor[52]. We expect that those approaches would perform particularly well in GHz-bandwidth regime and beyond. On the other hand, to enhance the resolution of our scheme to below 1 kHz a quantum memory with much longer lifetime would suffice, which has already been demonstrated in an optical lattice[53]. This lifetime does not necessarily change the super-resolution enhancement factor itself, which for the given temporal/spectral aperture only depends on the quality of interference and noise properties of the device.

Remarkably, the PuDTAI protocol has been realized here in a quantum memory that is insensitive to the signal spatial distribution, i.e., is spatially-multimode[54]. This means that it can accept external fluorescence light that is in principle spatially-multimode. Furthermore, we note that to analyze fluorescent light from different kinds of samples, one needs to match the wavelength of operation. This can be done via the maturing techniques of quantum frequency conversion[55].

The path from resolving two sources to obtaining a more complex spectrum with super-resolved features is challenging, yet some knowledge can be drawn from imaging problems considered so far. Various groups have considered the cases of three sources in two dimensions[56,57], or two sources with arbitrary brightness[58], and practical frameworks for multi-pixel images are being developed as well[59]. For the spectroscopic case discussed here, different operations in the quantum memory will be needed to prepare measurements optimal for multi-source scenarios.

## Methods

**Gradient echo memory.** The Gradient Echo Memory is based on pencil-shaped ($8 \times 0.3 \times 0.3$mm³) $^{87}$Rb atomic cloud prepared in magnetooptical trap. The full experimental sequence is depicted in Fig. 7a. After the cloud preparation stage (trapping, compression, and cooling) the atoms are optically pumped to the $|h\rangle = (F = 2, m_F = -2)$ state. The control field (795 nm) couples the excited state $|e\rangle = (F = 1, m_F = -1)$ with the previously emptied storage state $|g\rangle = (F = 1, m_F = 0)$ which enables coherent absorption of the signal field at the $|h\rangle \rightarrow |e\rangle$ transition (see Fig. 7b for reference). The magnetic field gradient crucial to the GEM protocol is generated by two identical coils located antisymmetrically on the $z$ axis and powered by fast H-topology current switch, that allows switching between negative and positive gradients within 5 μs. In the experiment we set the gradient to $\partial_z B_z = \pm$ 7.3 μT/mm Additionally, to separate the magnetically broadened absorption spectrum of the $|h\rangle \rightarrow |e\rangle \rightarrow |g\rangle$ transition from the unbroadened clock transition $(F = 2, m_F = -1) \rightarrow (F = 1, m_F = 0) \rightarrow (F = 1, m_F = 1)$, we keep the cloud in bias magnetic field of $B_z = 120$ μT magnitude along the $z$-axis. Moreover, to stabilize the two-photon detuning that is sensitive to magnetic fields we operate the protocol at 50 Hz synced with local mains frequency and correct for slow magnetic fluctuations (caused, for example, by an elevator near our laboratory) by adjusting the magnitude of the bias magnetic field in a feedback scheme described below. The temperature of the cloud is measured to be about 60 μK. Given the configuration of laser beams, i.e., angle between coupling and signal light, the atomic coherence has a 50 rad/mm transverse wavevector component. This gives ≈260 μs of thermal-motion-limited storage time[60]. The control field-induced spin-wave decay rate[37] in the experiment is about 10kHz.

The signal photons are passed through an optically-pump $^{87}$Rb-vapor cell in order to remove the residual control field. At the end, they are coupled to a single-mode fiber and detected with an SPCM.

**Magnetic field synchronization and stabilization.** The protocol operates at 50 Hz repetition rate synced with local mains frequency. The synchronization mechanism is realized by the FPGA (NI 7852R) system controlling the whole sequence. The synchronization (waiting for mains trigger) happens during the trapping period of each experimental repetition effectively providing no-idle-time operation. The slow (below 50 Hz) magnetic field fluctuations are compensated by feedback mechanism that is enabled at the end of each sequence repetition. This is achieved by repumping the atoms to the $m_F = -2$ sublevel, switching off the GEM gradient, and probing the atomic spin precession with linearly polarized probe beam at $(F = 2) \rightarrow (F = 1)$ transition illuminating the whole ensemble from the side. The polarization-rotation signal is registered by a differential photodiode and the precession frequency is estimated in real time by fitting a quadratic function around the maximum of registered signal Fourier spectrum. From this, the frequency-error signal is obtained and fed to PI (proportional-integral) controller that modulates the current in the $z$-direction compensation coils (see Fig. 2a for reference). With the compensation mechanism active, we achieve long-term rms stability of 130 Hz of the Larmor frequency, corresponding to 18.5 nT. This translates to the stability of the relative frequency centroid of twice the Larmor frequency, i.e., 260 Hz, which is well below the desired resolution.

**Signal pulse preparation.** The signal and control pulses are carved out from two branches of a continuous wave frequency-stabilized[61] laser. The first branch, used for signal pulse generation is derived by frequency-shifting by 6834 MHz part of laser light using EOM and Fabry–Pérot filtering cavity (see ref. [41] for details). Then the signal pulses are carved with double-pass AOMs feed by home-made arbitrary waveform generator consisting of *Red Pitaya* running the *PyRPL*[62] software and external frequency mixer providing upconversion of the carrier frequency to the desired 80 MHz. The setup is depicted in Fig. 7c. The *Red Pitaya* with *PyRPL*

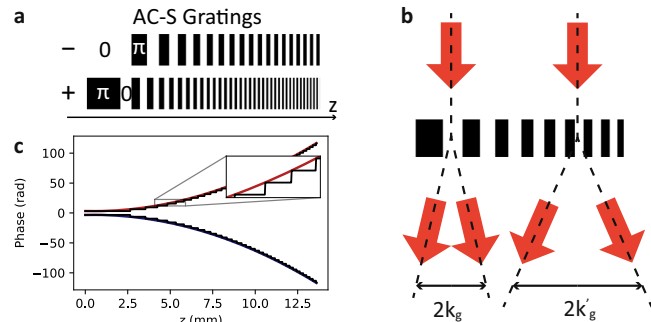

**Fig. 8 Chirped square-wave phase grating used to simultaneously apply positive and negative quadratic phase in space. a** Two gratings used to implement the antisymmetric (−) and symmetric (+) PuDTAI ports. **b** The coherence at longitudinal positions $z$ is split into diffraction orders that are separated by the grating local wavevector $k_g = \kappa z$. **c** Interpretation of the chirped square-wave grating as a superposition of positive and negative quadratic Fresnel phase profiles.

allows us to prepare arbitrary (up to bandwidth limitations of 50 MHz) complex envelopes that are internally multiplied by the carrier waveform at 35 MHz using digital IQ-mixer. The signal is then externally mixed with 45 MHz local oscillator from Direct Digital Synthesizer controlled by the main FPGA system to match the AOM central frequency of 80 MHz and provide extended amplitude dynamic range.

The frequency-domain double-Gaussian signal $\tilde{S}_\varphi(\omega)$ is generated by programming a temporal envelope $S_\varphi(t) = S \cos(\frac{\delta\omega t - \varphi}{2}) \exp(-t^2\sigma^2)$ with given $\delta\omega = \sigma\varepsilon$. During the measurements, to make the virtual sources mutually incoherent, we continuously (mod $2\pi$) change the phase $\varphi$ from $-\pi$ to $\pi$ in a way that for a single measurement the number of complete $2\pi$ cycles is in order of few thousands.

**PuDTAI model and calibrations.** The schematic representation of the theoretical model is depicted in Fig. 7d. The model follows image-inversion interferometer description with two mutually incoherent weak ($\bar{n} \ll 1$) sources at the input.

By design, our interferometer superimposes the two arms in the Fourier space. This can be most easily understood by investigating the Wigner function transformations of the two signal pulse halves during propagation through the interferometer. Let us focus on the first half that has a temporal amplitude:

$$\mathcal{A}_-(t) = \begin{cases} \mathcal{A}(t) & t < 0 \\ 0 & t \geq 0 \end{cases}. \tag{9}$$

The time-dependent two-photon detuning which for $t < 0$ is $\delta(t) = \alpha t$ virtually applies a temporal phase profile $\mathcal{A}_-(t) \rightarrow \mathcal{A}_-(t) \exp(i\frac{\alpha}{2}t^2)$ that corresponds to $z \rightarrow z' = z - \alpha k_z$ transformation of the associated atomic coherence Wigner function $\mathcal{W}_-(z, k_z)$. In the language of temporal imaging, this operation is known as propagation through temporal lens with the focal length $f_t = \omega_0/\alpha$[37] where $\omega_0$ is the optical carrier frequency. The SSM modulation with spatially chirped square grating with wavevector $k_g = \kappa z$ splits the coherence into multiple diffraction orders as depicted in Fig. 8. However, as the depth of the grating is $\pi$ the 0th diffraction order vanishes and the coherence is mostly split to ±1 orders (ideally, only 18% is distributed into higher orders). In the case of the first half of the signal pulse we are interested in the—1st order, that when isolated can be understood as a result of quadratic phase modulation in the real space: $\varrho_{hg}(z) \rightarrow \varrho_{hg}(z) \exp(i\frac{\kappa}{2}z^2)$. This corresponds to the $k_z \rightarrow k'_z = k_z \pm \kappa z$ transformation of the Wigner function $\mathcal{W}_-(z, k_z)$ and in terms of temporal imaging represents temporal propagation by a distance $d_t = \kappa/\omega_0\beta^2$ where $\beta$ is the Zeeman shift gradient slope that facilitates the spectrum-to-space mapping. With $\kappa = 1/\alpha$ that means $d_t = f_t$ the full transformation reads:

$$k_z \rightarrow k'_z = \kappa z \tag{10}$$

$$z \rightarrow z' = z - \frac{1}{\kappa} k_z \tag{11}$$

which up to the additional temporal phase modulation represents a counter-clockwise 90° rotation of the phase-space, that in fact is a backward Fourier transform. Similarly for the second half of the pulse (for $t \geq 0$), as the sign of $\alpha$ changes, we get a clockwise rotation and thus forward Fourier transform. Finally, as at the input we have the full signal pulse $\mathcal{A}(t) = \mathcal{A}_-(t) + \mathcal{A}_+(t)$ the two components interfere in the Fourier domain when the positive frequency components of the first half are superimposed onto the negative components of the second half thus implementing the inversion interferometer. Additionally, the

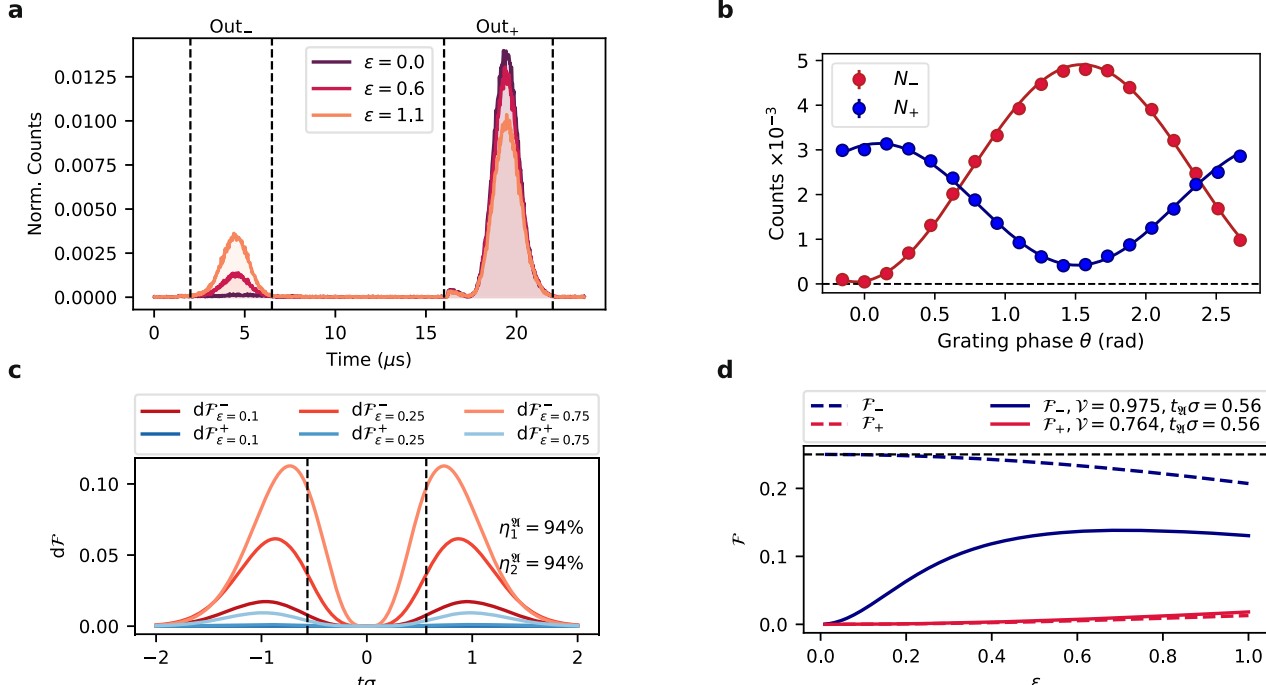

**Fig. 9 Time-bin outputs of the PuDTAI interferometer and associated FI. a** Single-photon counts distributions registered at each port (Out_, Out_+) of the PuDTAI interferometer for several chosen true separation values $\varepsilon$. **b** The PuDTAI interferometer phase $\theta$ scan for fixed separation parameter $\varepsilon = 0$ along with fitted sine and cosine functions corresponding to two output ports ($N_-$ and $N_+$). **c** Fisher information density $d\mathcal{F}_\varepsilon$ for direct detection of the antisymmetric ($-$) and symmetric ($+$) subspace with the realistic parameters $\mathcal{V}_-$ and $\mathcal{V}_+$. For small $\varepsilon$ the information is concentrated in the two lobes outside the central part of the spectrum. The relative efficiencies $\eta_-^\mathfrak{A}$, $\eta_+^\mathfrak{A}$ correspond to the hard aperture (marked as dashed lines) that removes the input signal located within the $t\sigma = \pm 0.564$ range. **d** Fisher information available at each output port of the PuDTAI device. For small separations $\varepsilon \ll 1$ the antisymmetric port ($\mathcal{F}_-$ corresponding to $p_-$) contains the most information about $\varepsilon$. The dashed lines represents the ideal case of $\mathcal{V}_\pm = 1$ and infinite aperture.

by-design Fourier transform feature of our PuDTAI interferometer can improve the signal-to-noise ratio when using noisy detectors as long signal pulses result in short pulses at the interferometer output.

With the separation parameter $\varepsilon = \delta\omega/\sigma$ and temporal representational of the input signal given by:

$$\mathcal{S}_\varphi(t) = \psi_\mathcal{G}(t)\sqrt{2}\cos\left(\frac{\delta\omega t - \varphi}{2}\right)\exp\left(\frac{i\varphi}{2}\right) \quad (12)$$

that at the interferometer input is clipped by the (symmetric) temporal aperture function $f_\mathfrak{A}(t)$, we may write the following amplitudes for the antisymmetric and symmetric ports:

$$u_-(t) = \frac{1}{2}f_\mathfrak{A}(t)(\mathcal{S}_\varphi(t) - \mathcal{S}_\varphi(-t)), \quad (13)$$

$$u_+(t) = \frac{1}{2}f_\mathfrak{A}(t)(\mathcal{S}_\varphi(t) + \mathcal{S}_\varphi(-t)). \quad (14)$$

As the interferometer internally performs a Fourier transform of the input signal, at the device output we observe the counts distributions given by these two components in the frequency-domain $\tilde{p}_i(\omega) = |\tilde{u}_i(\omega)|^2$ (see Fig. 9a for example of measured counts distributions, with $\omega = \alpha t$).

The total contributions in the antisymmetric and symmetric ports are calculated as $p_i = \int \tilde{p}_i(\omega)d\omega$, which corresponds to using a spectral bucket (no frequency information) detector. After introducing non-perfect interference visibilities $\mathcal{V}_-$ and $\mathcal{V}_+$ and additional losses in the symmetric port $\eta_+$ we arrive at outcome probabilities that for hard Cassegrain-type aperture function given by $f_\mathfrak{A}(t) = \Theta(-t_\mathfrak{A} - t) + \Theta(-t_\mathfrak{A} + t)$ take the form:

$$p_- = \frac{1}{2}\left(\text{erfc}(\sqrt{2}\sigma t_\mathfrak{A}) - \mathcal{V}_- e^{-\frac{\varepsilon^2}{8}}f(t_\mathfrak{A}, \varepsilon)\right), \quad (15)$$

$$p_+ = \frac{\eta_+}{2}\left(\text{erfc}(\sqrt{2}\sigma t_\mathfrak{A}) + \mathcal{V}_+ e^{-\frac{\varepsilon^2}{8}}f(t_\mathfrak{A}, \varepsilon)\right), \quad (16)$$

$$p_\times = 1 - p_- - p_+, \quad (17)$$

where $f(t_\mathfrak{A}, \varepsilon) = \frac{1}{2}(\text{erfc}(\frac{4t_\mathfrak{A}\sigma - i\varepsilon}{2\sqrt{2}}) + \text{erfc}(\frac{4t_\mathfrak{A}\sigma + i\varepsilon}{2\sqrt{2}}))$. The $p_-$, $p_+$ are probabilities of detecting photon in the antisymmetric and symmetric interferometer port

respectively, while $p_\times$ is a probability of no detection event. In Fig. 9b, we show the contributions to the symmetric and antisymmetric port for $\varepsilon = 0$ when varying the grating phase $\theta$, which demonstrates the high interference visibility.

The atomic cloud size along with the strength of the magnetic field gradient and slope of the control field chirp $\alpha$ limits the temporal aperture $\mathfrak{A}$ of the interferometer. We design the aperture to contain the most informationally valuable parts of input signal while at the same reducing the impact of the instantaneous control chirp reversal at $t = 0$. This reversal by being inherently broadband spoils the time-to-position mapping by virtually locally broadening the input signal. To get rid of this effect we chose the central two-photon detuning $\delta(t = 0)$ to be outside the magnetically broadened absorption spectrum providing no light-to-atoms mapping at the $\alpha$ reversal time. This forms an Cassegrain-type aperture described by $f_\mathfrak{A}$ that limits mostly the efficiency in the second interferometer port, that for $\varepsilon \ll 1$ contains no information, and serves only as brightness reference for the first port. We show this in Fig. 9c, where we plot the Fisher information densities defined as:

$$d\mathcal{F}_i = \frac{1}{p_i(t)}\left(\frac{\partial}{\partial\varepsilon}p_i(t)\right)^2 dt. \quad (18)$$

The impact of the aperture on photon flux efficiencies in the symmetric and antisymmetric port can be investigated by comparing the photon detection probabilities (Eqs. (15) and (16)) with the case of infinite aperture $t_\mathfrak{A} = 0$ and taking the limit of $\varepsilon \to 0$. This gives $\eta_{p_-}^\mathfrak{A} \approx \text{erfc}(\sqrt{2}\sigma t_\mathfrak{A}) + 2\sqrt{2/\pi}\sigma t_\mathfrak{A}\exp(-2\sigma^2 t_\mathfrak{A}^2)$ and $\eta_{p_+}^\mathfrak{A} \approx \text{erfc}(\sqrt{2}\sigma t_\mathfrak{A})$ which for realistic aperture $t_\mathfrak{A} = 0.564/\sigma$ gives $\eta_{p_-}^\mathfrak{A} \approx 0.74$, $\eta_{p_+}^\mathfrak{A} \approx 0.26$. However, in the context of $\varepsilon$ estimation those are not crucial, and we should look how the FI changes with the $t_\mathfrak{A}$. The FI density (18) integrated over the aperture gives the total FI available at the outputs of the device $\mathcal{F}_i = \int_\mathfrak{A} d\mathcal{F}_i$ that can be utilized when using a spectrally-resolved detection. Notably, in the case of infinite aperture and ideal visibility the total FI at the output $\mathcal{F}_\Sigma^\infty = \mathcal{F}_-^\infty + \mathcal{F}_+^\infty$ is independent of $\varepsilon$ and equals QFI which means that shapes of the distributions at each port of the device contain additional information that can improve the resolution for larger $\varepsilon$ values as has been previously recognized in the context of real-space imaging[22]. To investigate the impact of the finite aperture $\mathfrak{A}$, we calculate the FI $\mathcal{F}_\pm^\mathfrak{A}$ and evaluate the efficiencies as $\eta_{\mathcal{F}_\pm}^\mathfrak{A} = \mathcal{F}_\pm^\mathfrak{A}/\mathcal{F}_\pm^\infty$, which in the case of our aperture and visibilities both approximately equal 94% as included in Fig. 9c. In the case of spectral bucket detector, the total Fisher information

calculated for the observable outcome probabilities (($13$),($14$)) takes the form:

$$\mathcal{F}_{\mathrm{PuDTAI}} = \underbrace{\frac{\mathcal{V}_-^2 F}{64 p_-}}_{\mathcal{F}_-} + \underbrace{\frac{\eta_+^2 \mathcal{V}_+^2 F}{64 p_+}}_{\mathcal{F}_+}, \tag{19}$$

$$F = e^{-\varepsilon^2/4}\left(\sqrt{8/\pi}\, e^{-2t_{\mathfrak{A}}^2 \sigma^2 + \varepsilon^2/8}\, \sin(t_{\mathfrak{A}}\sigma\varepsilon) + \varepsilon f(t_{\mathfrak{A}},\varepsilon)\right)^2. \tag{20}$$

In Fig. 9d, we plot the two parts of $\mathcal{F}_{\mathrm{PuDTAI}}$ for both the ideal and realistic case.

Interestingly, when we now evaluate the FI efficiencies in a similar way as in the case of spectrally-resolved detection and with the finite visibilities we may observe improvement over the apertureless case for certain $\varepsilon$ range. This is a direct result of filtering the informationally not valuable central part of the signal that contains most of the leaking (due to $\mathcal{V} < 1$) photons. The improvement is most prominent for $\varepsilon \to 0$ and in our case reads $\eta_-^{\mathfrak{A}} = \eta_+^{\mathfrak{A}} \approx 2.1$. The experimental parameters $\mathcal{V}_- \approx (97.51 \pm 0.03)\%$, $\mathcal{V}_2 = (76.4 \pm 0.8)\%$, $t_{\mathfrak{A}}\sigma \approx 0.564 \pm 0.002$ and the efficiency $\eta_+ \approx 0.719 \pm 0.009$ are obtained from calibration measurements consisting on running the protocol for different $\varepsilon$ and interferometer phase $\theta$ as shown in Fig. 9b.

Finally, we can also evaluate the mean efficiencies of storage retrieval from the memory. We obtain $\eta_{p_-} = 0.83\%$ and $\eta_{p_+} = 0.60\%$. We note that those efficiencies are very similar to the ones obtained in the QPG protocol[24]. To boost the efficiencies of the GEM one could design a larger and more dense atomic ensemble or prepare the ensemble inside a cavity resonant with the signal field that effectively boosts the optical depth and thus the efficiency[63–65].

**Maximum-likelihood estimation.** The estimation of the separation parameter $\varepsilon$ in both cases (QMTI and PuDATI) follows the standard maximum-likelihood estimation procedure. In the case of QMTI, the single outcome probability function is given by

$$p_\varepsilon(\omega) = \tilde{I}(\omega) = \frac{1}{2}\left(|\tilde{\psi}(\omega - \sigma\varepsilon/2)|^2 + |\tilde{\psi}_-(\omega + \sigma\varepsilon/2)|^2\right). \tag{21}$$

For given distribution of measurement outcomes $\boldsymbol{\omega} = (\omega_1, ..., \omega_n)$ we construct the likelihood function

$$\mathcal{L}_{\boldsymbol{\omega}}(\varepsilon) = \prod_{i=1}^{n} p_\varepsilon(\omega_i), \tag{22}$$

that is numerically maximized to yield the maximum-likelihood value of $\varepsilon$. For PuDTAI, we have the three possible outcomes that are characterized by the three probabilities: $p_-, p_+, p_\times$. For $N$ processed photons we have the likelihood given by a probability mass function of the trinomial distribution:

$$\mathcal{L}_{N_-, N_+}(\varepsilon) = \frac{N!}{N_-! N_+! (N - N_- - N_+)!} \times \\ p_-(\varepsilon)^{N_-} p_+(\varepsilon)^{N_+} p_\times(\varepsilon)^{N - N_- - N_+}, \tag{23}$$

which is maximized for $p_-(\varepsilon)/p_+(\varepsilon) = N_-/N_+$. From this, we estimate the separation by numerically solving the formula for $\varepsilon$.

**Super-resolution parameter.** As given by Eq. ($7$), the super-resolution parameter $\mathfrak{s}$ must be evaluated at a vanishing separation $\varepsilon$. For the PuDTAI protocol, we directly plug in the formula for Fisher information (Eq. ($19$)) to the limit (Eq. ($7$)) and obtain:

$$\mathfrak{s}_{\mathrm{PuDTAI}} = \frac{e^{-4\sigma^2 t_{\mathfrak{A}}^2}\left(\mathcal{V}_-^2(\mathcal{V}_+ + 1) - \eta_+ \mathcal{V}_- \mathcal{V}_+^2 + \eta_+ \mathcal{V}_+^2\right)}{4(\mathcal{V}_- - 1)(\mathcal{V}_+ + 1)\left(\mathrm{erf}\left(\sqrt{2}\sigma t_{\mathfrak{A}}\right) - 1\right)} \\ \times \left(-e^{2\sigma^2 t_{\mathfrak{A}}^2}\mathrm{erf}(\sqrt{2}\sigma t_{\mathfrak{A}}) + e^{2\sigma^2 t_{\mathfrak{A}}^2} + 2\sqrt{\frac{2}{\pi}}\sigma t_{\mathfrak{A}}\right)^2 \tag{24}$$

The value may be evaluated accurately with an uncertainty, as we know all the relevant experimental parameters. Finally, if we set $\mathcal{V}_- = \mathcal{V}_+ = \mathcal{V}$ and $\eta_+ = 1$ and $t_{\mathfrak{A}} = 0$, we obtain $\mathfrak{s}_{\mathrm{PuDTAI}} = \mathfrak{s}_{\mathrm{SLIVER}}$ (Eq. ($8$)).

The classical (DI) spectrometers such as Grating, FT, and QMTI that are not tailored to the given signal mode function $\tilde{\psi}(\omega)$ can be directly used with signals that have different FWHM or $\sigma$. To indicate that, we represent them as lines covering some $\sigma$ range that depend on given spectrometer implementation. The super-resolution parameter $\mathfrak{s}$ in this case is calculated using the same formula as in the case of spectrometers tailored for the given $\tilde{\psi}(\omega)$ such as PuDTAI or QPG. Any DI spectrometer has its bandwidth (BWL) and resolution limit (RL). These are causing the $\mathfrak{s}$ parameter to drop below 1 for signals with bandwidth exceeding the BWL or that are narrower than RL. The limitation from the finite RL broadens the signal mode function $\tilde{\psi}(\omega)$ and thus spoils the separation estimation sensitivity by virtually making the $\varepsilon$ smaller. For a RL given by $\sigma_{\mathrm{RL}}$ the broadening effect can be calculated as $\sigma \to \sqrt{\sigma^2 + \sigma_{\mathrm{RL}}^2}$, which result in the $\varepsilon$ smaller by $\frac{\sigma}{\sqrt{\sigma^2 + \sigma_{\mathrm{RL}}^2}}$ factor. On the other hand, the BWL characterized by $\Sigma_{\mathrm{BWL}}$ cuts the informationally valuable tails of signals with $\sigma > \Sigma_{\mathrm{BWL}}$ that results in drop of the Fisher information given as $\mathcal{F}_{\mathrm{DI}} = \int_{\Sigma_{\mathrm{BWL}}} d\mathcal{F}_{\mathrm{DI}}$ with the $d\mathcal{F}_{\mathrm{DI}}$ being the DI Fisher information density and the integration goes through the whole available bandwidth. In Fig. 6, we plot several $\mathfrak{s}$

curves for various DI spectrometers. These are: Grating—a grating spectrometer with gratings of lengths ranging from 1 to 10 cm, with grating period of 1/1200 mm and $\Sigma_{\mathrm{BWL}} = 10^3 \times \sigma_{\mathrm{RL}}$; FT—Bruker IFS 125HR Fourier transform spectrometer with 0.001 cm$^{-1}$ resolution and $50 \times 10^3$ cm$^{-1}$ and the MIR spectral range; QMTI —the Quantum Memory Temporal Imaging Spectrometer with $\sigma_{\mathrm{RL}} = 7.2$ kHz and 300 kHz BWL.

## Data availability

Data presented in Fig. 5 and Fig. 9 have been deposited in RepOD Repository for Open Data at https://doi.org/10.18150/JSUKE9. Any other data that support the findings of this study are available from the corresponding author upon reasonable request.

## Code availability

Code used in data analysis is available from the corresponding author upon reasonable request.

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

## Acknowledgements

We thank W. Wasilewski, M. Jachura, and R. Demkowicz-Dobrzański for insightful discussions and K. Banaszek for the generous support. This work has been funded by National Science Centre (Poland) grant no. 2017/25/N/ST2/00713, Polish science budget funds for years 2017-2021 as a research project within the "Diamentowy Grant" program of the Ministry of Science and Education (DI2016 014846), Office of Naval Research (USA) grant no. N62909-19-1-2127 and by the MAB/2018/4 "Quantum Optical Technologies" project. The Quantum Optical Technologies project is carried out within the International Research Agendas program of the Foundation for Polish Science co-financed by the European Union under the European Regional Development Fund. Michał Parniak and Mateusz Mazelanik were also supported by the Foundation for Polish Science via the START scholarship.

## Author contributions

M.M. and M.P. conceived the scheme and planned the experiment and the theoretical research. M.M., A.L., and M.P. contributed to the experimental setup, software, and data collection. M.M. analyzed the data, developed theory, and prepared figures. M.M. and M.P. wrote the manuscript. All authors discussed the results.

## Competing interests

The authors declare no competing interests.

## Additional information

**Peer review information** *Nature Communications* thanks Erhan Saglamyurek and the other anonymous reviewer(s) for their contribution to the peer review this work. Peer reviewer reports are available.

