## [Peer Review File · Nature Communications]

Reviewers' Comments:

Reviewer #1:

Remarks to the Author:

The manuscript "Optical-domain spectral super-resolution enabled by a quantum memory" by Mazelanik et al. is a very interesting manuscript, pertaining to the topic of super-resolution. Whereas super-resolution has become an established topic in imaging optics the novelty in this work is to apply similar techniques to the frequency domain.

In imaging optics one of the key concepts is the smallest spot size than can be achieved with a given system, and the unfortunate notation of "diffraction limit" is used to describe a certain limit. However, there are numerous schemes to beat this so-called limit, all of which use the physics of diffraction, therefore it is a topic that generates much debate. What is missing in the literature often is a discussion of signal to noise. When two point spread functions overlap, the central question is what displacement, as a fraction of the width can be resolved as 2 rather than 1 objects? Without a signal to noise aspect the discussion is likely to be invalid.

Similarly, when looking at two spectral lines closely spaced in frequency through a dispersive optical, there is a standard textbook criterion named after Lord Rayleigh, which essentially states that the separation has to exceed the width for two lines to be resolved. This is a rather naïve treatment, as it assumes both lines have the same brightness, and completely ignores signal to noise. Therefore it is very reassuring to see that the treatment given here build in concepts such as Fisher information.

Despite finding this a fascinating topic, and I believe the results obtained are very impressive, I found it very difficult to follow the manuscript. There are two main reasons for this: (i) non-standard English phrases are used, with missing articles etc. The manuscript would benefit enormously from being proof read more thoroughly. (ii) not enough effort has been made to explain the concepts presented here to a more generalised audience.

Recommendations.

General

A drastic editing of the manuscript is needed. More figures are needed early on to precisely define the term and concepts used, such that non-specialists can appreciate better what has been done. I would suggest some images of overlapping spectral lines of different signal to noise and different separation would greatly help with introducing the key concepts sequentially. Currently, figure 1 shows a schematic of the apparatus, and has two lines of infinite signal to noise. Then there is a lot of material to absorb before figure 2 appears, and by that stage it is not clear what is being plotted, or why they are the concepts of interest.

Specific comments on the introduction.

Page 1 A reference is needed for the "Rayleigh curse".

Page 1 In the paragraph stating that there is a super-resolution enhancement factor of 20.05 it would be good to state clearly how this enhancement is obtained; on what parameters does it depend; is there scope (in a different system) for an even higher enhancement.

Page 1 I do not follow the following statement, which is key to the whole technique. "performance is enabled by the long coherence time of the spin-wave quantum memory which serves to extract the quantum information from the optical field in an optimal way." It needs unpacking, and more explanation. Possibly even numerical values.

Reviewer #2:

Remarks to the Author:

In this manuscript, the authors develop and present an experimental implementation of a new spectral super-resolution method that is based on quantum metrology. To the best of my knowledge this is the first realization of such a quantum super-resolution protocol for spectroscopy. Quantum super-resolution ideas were developed and implemented recently in optical imaging, however their relevance to spectroscopy was not fully clear. A few theoretical

papers have been published on this issue, but so far no experimental realization has shown this effect. In fact, some issues (like the need of high detection fidelity) seemed to limit the applicability of these schemes. In this manuscript the authors tackle this problem from a different angle and find a clever and practical way to map the spatial super-resolution protocol to the spectral one. The method they develop, PuDTAI, is basically a spectral analog of the SLIVER method in imaging. This method relies on the use of quantum memory (specifically: atomic gradient echo memory) which enables performing a specific measurement that doesn't suffer from a vanishing Fisher information.

I find the ideas and the results quite exciting, and judging only according to them I believe the paper is suitable for publication in Nat. Comm. On the other hand the presentation and the explanations seem to be insufficient in some parts and a bit sloppy. Hence I'd be happy to recommend on publication given that these points are addressed:

1. It was hard and confusing for me to understand the mapping of the spectrum to the quantum memory and the details behind each operation. In fig. 1a you present the method where each ingredient is a black box.

I was hoping to see a table or a figure that clarifies how each operation is being performed using the GEM.

Currently it's a bit messy (and the discussion about the Wigner function only added to my confusion).

It would be great if you first put a table with each ingredient and what's the actual implementation and afterwards explain

why each operation does what it does.

Fig. 2 (b) is also hardly explained (it's just impossible to understand it from the captions):

In the first layer of (b):

what does the red and the blue stand for? What's the right column and what's the left column? Does red mean positive and blue negative? Does the right column correspond to antisymmetric and the left column to symmetric?

Please clarify all these things.

2. (a) The generality/ applicability of the scheme to different spectroscopy problems is unclear.

In your experiment, the width of the signal was generated artificially, however in most actual problems this width stems from either limited sampling/interrogation time or noisy amplitudes/phases. It seems that the scheme should work for finite width due to finite interrogation time (?), but what about the case of noisy amplitudes/phases?

If the amplitudes undergo a noise process, then these amplitude/phase fluctuations should ruin the symmetry between the two halves. Then basically even if there is a single frequency you'd get a finite probability for the antisymmetric function.

Hence it seems that noisiness of the signal may still impose a limitation. Is that right?

Can you maybe elaborate more on the applicability of the method to actual signals and the implications of noise?

(b) I also wasn't able to understand what's the fidelity of mapping the spectrum to the memory and if imperfect fidelity of this mapping should affect the process?

3. Typos and minor comments:

(a) In general it would be better if you use more the terminology of spectroscopy-you use Rayleigh limit, aperture which are all taken from imaging.

In spectroscopy you usually sample a signal and then the limit is basically the Fourier limit (determined by the total sampling time/ intrinsic noise of the signal).

Can you also define exactly what's the temporal aperture.

(b) p. 2 last paragraph: "...manipulations in a an..."

(c) Same paragraph: $\rho_{\{hg\}}(z)$ is mentioned for the first time and is undefined, please define

(d) In p.3 third paragraph suddenly appears $\rho(z)$, what's the definition?

(e) P. 3 last paragraph of the first column "The chirped grating grating..."

(f) P. 4 above "Experimrntal Results" it should be "also known" instead of "also know"

(g) P. 9 first column "to to"

(h) P. 9 last paragraph should be "propagation through" instead of "propagation though"

(i) Captions of fig. 10, should be " $\sigma = \pm 0.564$ "?

Reviewer #3:

Remarks to the Author:

In this paper, M. Mazelanik et al. develop and experimentally demonstrate a spectral-domain super-resolution technique using the advanced signal processing-capabilities of the Gradient-echo memory (GEM) light-storage protocol. While this technique (referred to as PuDTAI) is inspired by another super-resolution approach (SLIVER, originally proposed for the spatial domain), its realization via a cold-atom GEM offers a narrow-band and single-photon level operation together with a high-resolution capability (kHz range in the optical domain), which has not been achieved by any other spectral super-resolution scheme.

To demonstrate a resolution-enhancement with PuDTAI, the authors also utilize their quantum memory for implementing a direct imaging (DI) technique (i.e, direct spectral measurement), whose resolution is fundamentally limited by the Rayleigh criterion. Their comparative analyses of the experimental data (in terms of the precision of the frequency resolution) show that the performance of PuDTAI surpasses not only that of the experimental DI, but also overcomes the theoretical bound of a DI (imposed by the Rayleigh limit). These results, supported by the authors' further analysis, clearly demonstrate the super-resolution enhancement of the proposed technique. While a similar experimental accomplishment was previously reported in another study that uses a different super-resolution approach (Ref. 24), as a distinct feature, the current study achieves it in a vastly different operation-bandwidth and resolution-range.

In my opinion, this study brings several novelties and the reported progress is substantial. It opens up a new avenue by introducing a quantum-memory implementation as an advanced spectroscopic tool surpassing the Rayleigh limit. In this respect, to the best of my knowledge, the reported experiment is the first one that demonstrates a real advantage of the signal-processing capability of atomic quantum memories over conventional interferometry. Furthermore, the proposed technique features an operation regime with an ultra-low input-light level and a high-resolution, that is not accessible to the known spectral super-resolution techniques, and hence may find unique applications in spectroscopy, interferometry and quantum information processing. Finally, I point out that this manuscript presents a comprehensive performance and benchmark analyses, which can be a really good reference and guide for future studies on spectral super-resolution.

On the other hand, I have an impression that the current form of the manuscript is not easily accessible to a broad range of audience targeted by Nature Communications due to the level of the technical language and presentation style. Despite the author's effort towards simplifying things with nice visual aides and repeated explanations, I think that it is difficult to follow some part of the paper without a solid background in Gradient-echo memory, super-resolution techniques, and

statistical analysis.

In particular, the GEM protocol, which is at the heart of the proposed technique, could have been described in a simpler language, given that the basic photon-echo mechanism behind this approach provides an excellent opportunity to do so. However, instead of taking this opportunity for an intuitive physical insight into the implementation of GEM-based PuDTAI, the authors follow an abstract, yet concise, way of describing it (like rotation of the Wigner function). In this situation, specifically, I suggest the authors to add a figure, showing the spin-coherence evolution/distribution in the basic GEM implementation (without the temporal and spatial phase modulation). This figure could be presented before Fig.2 (and perhaps as a part of Fig. 1), serving as a guide for readers for better understanding the advanced steps of the implemented technique. Also, the basic link between the spatial and temporal phase terms of spin coherence (kz and ωt , respectively) should be shown in a simple manner, which is in the essence of the GEM protocol. This explicit connection would help a non-specialist in gaining better insight into why and how k (wave-vector) evolves when the two-steps phase modulations are realized. I also note that the manuscript attempts to describe the GEM-based implementation of PuDTAI in a couple of occasions towards making it more understandable. However, I feel that the totality of these descriptions (almost in the same style, but with slightly different wording) does not serve for the intended purpose. Beside expressing the associated transformations, at least in one of these repeated parts, one could also explain the basic intuitive connection between the temporal-phase modulation and stored frequency components of the signal as well as between the spatial-phase modulation and the frequency distribution of the stored spin-wave due to the imposed AC-Stark shift.

Similarly, decomposing an input waveform into symmetric and anti-symmetric components is the key idea of the PuDTAI interferometer, which is already mentioned in the beginning of the paper, but the reason and intuition behind this principle is not obvious to a reader who is non-specialist in super-resolution spectroscopy. Actually, just in the middle of the paper, the authors describe this fact clearly with a simple explanation and basic mathematical expressions (page-4, Eqns 4,5). In this case, I suggest the authors to introduce this part before the description of the PuDTAI technique, which would put everything in a more transparent context for easy understanding.

As mentioned above, the presented analyses of the experimental results are very systematic and informative. On the other hand, the authors could use a bit more "introductory language" when mentioning the basic concepts, including the Rayleigh curse, Cramer-Rao bound, and Fisher Information. In addition, the calculated/measured basic statistical parameters (e.g., max. likelihood estimator, variance) should be defined and recapped in such way that the results presented in 3a-c should give a clear insight into what is achieved in this study. For example, according to Fig. 3a, the raw estimates of the spectral separation parameter (ϵ) yield the same value for QMTI and PuDTAI, but with vastly different uncertainty. The implication of this result needs a bit more clarification in practical terms (e.g., how this parameter indicates the superiority of PuDTAI over QMTI in a certain measurement setting).

In short, I think that this study by M. Mazelanik et al meets the essential publication-criteria of Nature Communications from the aspects of scientific quality, technical advance and potential impact. However, I believe that the manuscript should be more accessible to a general audience, considering the multi-disciplinary nature of this journal. In this regard, following a potential revision that addresses the above points, I would be happy to recommend this article for publication in Nature Communications.

In the revised version, I also would like the authors to address the following minor comments and points:

1. In the introduction, the authors mention that long coherence-time of spin-wave memory is an important factor to extract the relevant spectral information from light in an optimal way. I would like the authors elaborate on this fact (for example, in discussion section), expressing how the longer storage times would boost the performance of the current PuDTAI implementation.
2. The authors often use the terminology of the space-domain imaging for explaining the

frequency domain operations. In this respect, I think that the analogy between a spatial aperture and spectral aperture (controlled by quantum memory) is not clearly described (on page-2, right column, first paragraph). I would suggest authors to make a simple imaging diagram (similar to Fig. 2a), showing to one-to-one correspondence of the relevant terms (e.g imaging, focusing etc.) between the two domains.

3. Fig 1a is a really nice illustration for describing time-inversion interferometry, but Fig 1b needs a bit more explanation (perhaps in the caption). Especially, the real function of the GEM coils is not described until Method section. Also, it took a while for me to interpret what the blue and red curves flying through the coils. So, they should be specified. In addition, there should be a color-scale for the plots shown in Fig. 2b

4. Fig 3a seems to make an analogy between the space and time/frequency domain pictures of a time-inversion interferometer. But there is no explanation of what each image-piece and labeling represent and refer to. Especially, it was a bit a puzzle for me to understand the meanings of the colored curves (inside the pulse envelopes), the pair of blue objects in the middle, and labels of the interferometer's inputs (until I saw a similar figure (fig.7) with a bit more explanation in Methods).

5. I think that the timing sequence, illustrated in Fig-5, is quite useful for understanding what is going on with the experimental implementation. The authors may consider moving it (or its simplified version) to the main text.

Reviewer #1:

The manuscript “Optical-domain spectral super-resolution enabled by a quantum memory” by Mazelanik et al. is a very interesting manuscript, pertaining to the topic of super-resolution. Whereas super-resolution has become an established topic in imaging optics the novelty in this work is to apply similar techniques to the frequency domain.

In imaging optics one of the key concepts is the smallest spot size than can be achieved with a given system, and the unfortunate notation of “diffraction limit” is used to describe a certain limit. However, there are numerous schemes to beat this so-called limit, all of which use the physics of diffraction, therefore it is a topic that generates much debate. What is missing in the literature often is a discussion of signal to noise. When two point spread functions overlap, the central question is what displacement, as a fraction of the width can be resolved as 2 rather than 1 objects? Without a signal to noise aspect the discussion is likely to be invalid.

Similarly, when looking at two spectral lines closely spaced in frequency through a dispersive optical, there is a standard textbook criterion named after Lord Rayleigh, which essentially states that the separation has to exceed the width for two lines to be resolved. This is a rather naïve treatment, as it assumes both lines have the same brightness, and completely ignores signal to noise. Therefore it is very reassuring to see that the treatment given here build in concepts such as Fisher information.

Despite finding this a fascinating topic, and I believe the results obtained are very impressive, I found it very difficult to follow the manuscript. There are two main reasons for this: (i) non-standard English phrases are used, with missing articles etc. The manuscript would benefit enormously from being proof read more thoroughly. (ii) not enough effort has been made to explain the concepts presented here to a more generalised audience.

Recommendations.

General

A drastic editing of the manuscript is needed. More figures are needed early on to precisely define the term and concepts used, such that non-specialists can appreciate better what has been done. I would suggest some images of overlapping spectral lines of different signal to noise and different separation would greatly help with introducing the key concepts sequentially. Currently, figure 1 shows a schematic of the apparatus, and has two lines of infinite signal to noise. Then there is a lot of material to absorb before figure 2 appears, and by that stage it is not clear what is being plotted, or why they are the concepts of interest.

We thank the Reviewer for appreciating our work and finding the results impressive. Moreover, we are grateful for the constructive criticism and the advice on how to improve the manuscript and make it accessible for a more general audience. Following Reviewers suggestions we introduced an additional section in the manuscript called “Temporal Imaging in GEM”. The section includes explanations of the basic concepts of temporal imaging and explains how each component is implemented in the quantum

memory. We also provide an additional figure with visualizations of the explanations arranged in the step-by-step manner. We hope that with this section the implementation of the PuDTAI interferometer is easier to follow. Moreover, we carefully proofread the manuscript and corrected language errors.

Specific comments on the introduction.

Page 1 A reference is needed for the “Rayleigh curse”.

We added the reference for the “Rayleigh curse” in the place where it is mentioned for the first time in the manuscript. The “Information in spectral resolution” section now also has more argumentation for using this peculiar terminology.

Page 1 In the paragraph stating that there is a super-resolution enhancement factor of 20.05 it would be good to state clearly how this enhancement is obtained; on what parameters does it depend; is there scope (in a different system) for an even higher enhancement.

The enhancement factor is obtained by taking the zero separation limit of the ratio of PuDTAI Fisher Information (FI) formula for realistic experimental parameters to the FI formula for perfect, noiseless direct imaging (DI) as explained in the Discussion section and defined by Eq. 7. We have now extended this discussion significantly. The basic interpretation of such parameter is how many times less photons are sufficient to obtain the same resolution (i.e. variance of estimator of separation) as the DI approach, in the limit of small separations. The improvement is mainly limited by the interference visibility and scales roughly as $V^2/(1 - V^2)$, which we now state in Eq. 8. In the Methods section we also give the full formula for our PuDTAI protocol. The visibility parameter takes into account noise and imperfections of the PuDTAI device implementation. In our case, the main limitation comes from noisy detection, and the spurious noise from the non-perfectly filtered coupling field. The detection noise could be improved by replacing the APDs with superconducting-nanowire photodetectors that have almost no dark counts. The filtering of the coupling field could be improved by employing a narrow filtering cavity resonant with the signal photons. On the other hand one could lower the power of the coupling field when a medium with higher optical density is used. The other imperfections include inhomogeneities of magnetic field gradient and the AC-Stark beam, but those in our case have minor contribution to the visibility.

We predict that in much different systems the enhancement will still depend on similar parameters, i.e. modal cross-talks, visibilities and dark counts.

Page 1 I do not follow the following statement, which is key to the whole technique. “performance is enabled by the long coherence time of the spin-wave quantum memory which serves to extract the quantum information from the optical field in an optimal way.” It needs unpacking, and more explanation. Possibly even numerical values.

The message that we wanted to convey with this sentence is following: with a quantum memory we can capture the optical signal to be measured without loss of information. Then,

during the storage time we can apply a specific linear transformation on the stored field and thus realize the optimal measurement for the given estimation task. Here, the transformation corresponds to decomposing the field into symmetric and antisymmetric components in the spectral domain. The two components are then detected which turns out to be optimal measurement for the frequency separation estimation problem. The coherence time of the memory has to be sufficient to allow storage of the long optical pulse as well as to allow performing the linear operation. Longer storage time means that we can store longer pulses - the temporal aperture can be larger - and thus we can measure signals with narrower bandwidth. Please also see the answer to a similar question posed by Reviewer 3.

Following the Reviewer suggestion we divided the sentence and made the message more straightforward. We hope that the current form of this introductory part is more clear and easier to follow. The information about the coherence time has been included in the Methods section.

Reviewer #2:

In this manuscript, the authors develop and present an experimental implementation of a new spectral super-resolution method that is based on quantum metrology. To the best of my knowledge this is the first realization of such a quantum super-resolution protocol for spectroscopy. Quantum super-resolution ideas were developed and implemented recently in optical imaging, however their relevance to spectroscopy was not fully clear. A few theoretical papers have been published on this issue, but so far no experimental realization has shown this effect.

In fact, some issues (like the need of high detection fidelity) seemed to limit the applicability of these schemes.

In this manuscript the authors tackle this problem from a different angle and find a clever and practical way to map the spatial super-resolution protocol to the spectral one. The method they develop, PuDTAI, is basically a spectral analog of the SLIVER method in imaging.

This method relies on the use of quantum memory (specifically: atomic gradient echo memory) which enables performing a specific measurement that doesn't suffer from a vanishing Fisher information.

I find the ideas and the results quite exciting, and judging only according to them I believe the paper is suitable for publication in Nat. Comm.

On the other hand the presentation and the explanations seem to be insufficient in some parts and a bit sloppy. Hence I'd be happy to recommend on publication given that these points are addressed:

1. It was hard and confusing for me to understand the mapping of the spectrum to the quantum memory and the details behind each operation. In fig. 1a you present the method where each ingredient is a black box.

I was hoping to see a table or a figure that clarifies how each operation is being performed using the GEM.

Currently it's a bit messy (and the discussion about the Wigner function only added to my confusion).

It would be great if you first put a table with each ingredient and what's the actual implementation and afterwards explain why each operation does what it does.

We are glad to hear that our ideas as well as the obtained results have been found exciting and worthy of publication in the *Nature Communications* journal. We thank the Reviewer for the insightful comments and suggestions that we diligently address below.

Firstly, as suggested by the Reviewer we now provide an additional section that introduces the basic concepts of temporal imaging in GEM. The section includes a figure with examples and visualizations for each crucial step needed to implement basic components of the temporal imaging technique in GEM. We start with explanations of the direct imaging spectrometer that is implemented in the memory. For each step we give relevant transformation of the Wigner distribution and provide an example with annotations marking the corresponding evolution of the phase space. On that basis we introduce the PuDTAI protocol in the next section.

Secondly, the Figure 2 (now Figure 4) representing the Wigner distribution has been extended as well - each transformation is now represented using arrows as in case of DI approach. Moreover we now provide a table with a synopsis of the protocol implementation that explicitly defines each operation and links it with the transformation of the Wigner space.

Fig. 2 (b) is also hardly explained (it's just impossible to understand it from the captions):

In the first layer of (b):

what does the red and the blue stand for? What's the right column and what's the left column?

Does red mean positive and blue negative? Does the right column correspond to antisymmetric and the left column to symmetric?

Please clarify all these things.

As mentioned above, the Figure 3 has been updated following the Reviewers comments. The red/blue colors were used to mark the positive/negative parts of the pulse respectively. In the new version of the figure the phase of the input/output pulses is represented by the color according to the colorbar at the bottom. The columns are now explicitly entitled as "Symmetric" and "Antisymmetric". We hope that the updated figure is now clear and straightforward to follow.

2. (a) The generality/ applicability of the scheme to different spectroscopy problems is unclear.

In your experiment, the width of the signal was generated artificially, however in most actual problems this width stems from either limited sampling/interrogation time or noisy amplitudes/ phases. It seems that the scheme should work for finite width due to finite interrogation time (?),

but what about the case of noisy amplitudes/phases?

If the amplitudes undergo a noise process, then these amplitude/phase fluctuations should ruin the symmetry between the two halves. Then basically even if there is a single frequency you'd get a finite probability for the antisymmetric function.

Hence it seems that noisiness of the signal may still impose a limitation. Is that right? Can you maybe elaborate more on the applicability of the method to actual signals and the implications of noise?

We thank for this insightful question. It indeed provides an area for a broad discussion. In our scheme we may control the temporal aperture via shaping of the control field, or even more simply by using an intensity modulator such as AOM in front of the quantum memory. Therefore, the scheme is indeed naturally well-suited for signals with well-defined temporal structure (such as, for example, a square pulse with finite width). At the same time, resilience to noise (or lack thereof) may be understood in several ways. Some of them have been studied theoretically. For example, photon noise due to thermal statistics provides the same results for $\bar{n} \ll 1$ as coherent states. In general, we may also consider that both spectral lines would have unequal intensities. Indeed, this causes the first moment to appear irrespective of the separation. It has been shown that superresolution is preserved in this case. However, higher-order measurements are required, which we expect will be an interesting subject for future work. It is best to actually consider moment estimation. It is a valid approach as measuring all moments precisely will constitute an image (or spectrum, in our case). Our method, in its current state, provides an enhanced measurement of the first moment.

A separate topic is classical fluctuations of the system, which may limit the effective superresolution improvement. If the fluctuations are much slower than the pulse width, similarly as for thermal statistics our scheme will be unaffected. However, if the fluctuations of phase or amplitude are random during the pulse, then unfortunately this will lead to reduced visibility of interference. However, one could consider such fluctuations as a fundamental limitation, i.e. they will reduce the maximum information that can be extracted from the field. Finding an optimal scheme in the presence of such noise could be an excellent path for future studies.

As a concluding remark, we notice that the reduced visibility of interference is already considered mathematically in the Methods section.

We have added comments about the influence of fluctuations in the Discussion.

(b) I also wasn't able to understand what's the fidelity of mapping the spectrum to the memory and if imperfect fidelity of this mapping should affect the process?

The fidelity of the pure spectrum to position mapping is limited mainly by the width of the two-photon absorption spectrum, which is broadened by the weak control field to about 10 kHz FWHM. However, as we are implementing the time lens during the light-to-atoms mapping process, the spectrum of the signal pulse is broadened to match the magnetically broadened absorption spectrum of the whole cloud. For the used value of the magnetic field gradient the bandwidth of the cloud is about $2\pi \times 600$ kHz. This gives the time-bandwidth product of $2\pi \times 60$, which can be understood as a number of bins into which the spectrum of the signal pulse is divided. The other limiting factor is the linearity of the magnetic field gradient. However for small deviations from the true linear gradient, this can be compensated by adjusting the AC-Stark beam profile. In conclusion, we are working in a

regime of large time-bandwidth product that is more than sufficient to allow good performance of the protocol. In fact the visibility is limited by other factors such as the spurious noise present at the output.

3. Typos and minor comments:

- (a) In general it would be better if you use more the terminology of spectroscopy-you use Rayleigh limit, aperture which are all taken from imaging.
In spectroscopy you usually sample a signal and then the limit is basically the Fourier limit (determined by the total sampling time/ intrinsic noise of the signal).

We thank the reviewer for pointing out this inconsistency. We have now made sure that terminology is consistent and more appropriate. With this, we would like to point out that since we want to use analogies to imaging, we still use the Rayleigh limit terminology in some places.

Our terminology is now as follows. We introduce the Rayleigh limit as a starting point for the discussion about resolution. We then explain that in modern understanding, the Rayleigh's rule of thumb is not an appropriate method to determine resolution of an imaging device as it does not take into account the number of photons used.

The more accurate and appropriate treatment is to calculate the Fisher information and use the Cramer-Rao bound to obtain the true resolution limit that depends on the number of used resources (photons). By analogy, the conventional Fourier limit has to be reformulated to truly quantify the spectral resolution of a spectrometer. With this, we need to effectively redefine the Fourier limit by calling a spectrometer **Fourier-limited** when its sensitivity to the small separation parameter ϵ is bounded by the formula obtained for direct imaging approach, in a full analogy to the Rayleigh limit in imaging. This remains consistent with the terminology used in imaging. Furthermore, it also remains consistent with spectroscopic methods, as traditional devices such as grating or Fourier-transform spectrometers will be Fourier-limited in a perfect case. Within this understanding, our method outperforms such devices and therefore achieves performance beyond the **Fourier limit**.

Can you also define exactly what's the temporal aperture.

The meaning of the temporal aperture is now explained with the help of the additional figure (Fig 3b) representing the temporal imaging setup. In the language of spectro-temporal domain the temporal aperture is just the shape of the temporal detection window, in other words it limits the interrogation time.

- (b) p. 2 last paragraph: "...manipulations in a an..."

Corrected.

- (c) Same paragraph: $\rho_{\{hg\}}(z)$ is mentioned for the first time and is undefined, please define

The $\rho_{\{hg\}}(z)$ is now defined in this paragraph.

(d) In p.3 third paragraph suddenly appears $\rho(z)$, what's the definition?

The omitted $\{hg\}$ index was a mistake which has been corrected.

(e) P. 3 last paragraph of the first column "The chirped grating grating..."

Corrected.

(f) P. 4 above "Experimrntal Results" it should be "also known" instead of "also know"

Corrected.

(g) P. 9 first column "to to"

Corrected.

(h) P. 9 last paragraph should be "propagation through" instead of "propagation though"

Corrected.

(i) Captions of fig. 10, should be " $\sigma = \pm 0.564$ "?

Indeed. Corrected.

Reviewer #3:

In this paper, M. Mazelanik et al. develop and experimentally demonstrate a spectral-domain super-resolution technique using the advanced signal processing-capabilities of the Gradient-echo memory (GEM) light-storage protocol. While this technique (referred to as PuDTAI) is inspired by another super-resolution approach (SLIVER, originally proposed for the spatial domain), its realization via a cold-atom GEM offers a narrow-band and single-photon level operation together with a high-resolution capability (kHz range in the optical domain), which has not been achieved by any other spectral super-resolution scheme.

To demonstrate a resolution-enhancement with PuDTAI, the authors also utilize their quantum memory for implementing a direct imaging (DI) technique (i.e, direct spectral measurement), whose resolution is fundamentally limited by the Rayleigh criterion. Their comparative analyses of the experimental data (in terms of the precision of the frequency resolution) show that the performance of PuDTAI surpasses not only that of the experimental DI, but also overcomes the theoretical bound of a DI (imposed by the Rayleigh limit). These results, supported by the authors' further analysis, clearly demonstrate the super-resolution enhancement of the proposed technique. While a similar experimental accomplishment was previously reported in another study that uses a different super-resolution approach (Ref. 24), as a distinct feature, the current study achieves it in a vastly different operation-bandwidth and resolution-range.

In my opinion, this study brings several novelties and the reported progress is substantial. It opens up a new avenue by introducing a quantum-memory implementation as an advanced spectroscopic tool surpassing the Rayleigh limit. In this respect, to the best of my knowledge, the reported experiment is the first one that demonstrates a real advantage of the signal-processing capability of atomic quantum memories over conventional interferometry. Furthermore, the proposed technique features an operation regime with an ultra-low input-light level and a high-resolution, that is not accessible to the known spectral super-resolution techniques, and hence may find unique applications in spectroscopy, interferometry and quantum information processing. Finally, I point out that this manuscript presents a comprehensive performance and benchmark analyses, which can be a really good reference and guide for future studies on spectral super-resolution.

On the other hand, I have an impression that the current form of the manuscript is not easily accessible to a broad range of audience targeted by Nature Communications due to the level of the technical language and presentation style. Despite the author's effort towards simplifying things with nice visual aides and repeated explanations, I think that it is difficult to follow some part of the paper without a solid background in Gradient-echo memory, super-resolution techniques, and statistical analysis.

In particular, the GEM protocol, which is at the heart of the proposed technique, could have been described in a simpler language, given that the basic photon-echo mechanism behind this approach provides an excellent opportunity to do so. However, instead of taking this opportunity for an intuitive physical insight into the implementation of GEM-based PuDTAI, the authors follow an abstract, yet concise, way of describing it (like rotation of the Wigner function). In this situation, specifically, I suggest the authors to add a figure, showing the spin-coherence evolution/distribution in the basic GEM implementation (without the temporal and spatial phase modulation). This figure could be presented before Fig.2 (and perhaps as a part of Fig. 1), serving as a guide for readers for better understanding the advanced steps of the implemented technique. Also, the basic link between the spatial and temporal phase terms of spin coherence (kz and ωt , respectively) should be shown in a simple manner, which is in the essence of the GEM protocol. This explicit connection would help a non-specialist in gaining better insight into why and how k (wave-vector) evolves when the two-steps phase modulations are realized. I also note that the manuscript attempts to describe the GEM-based implementation of PuDTAI in a couple of occasions towards making it more understandable. However, I feel that the totality of these descriptions (almost in the same style, but with slightly different wording) does not serve for the indented purpose. Beside expressing the associated transformations, at least in one of these repeated parts, one could also explain the basic intuitive connection between the temporal-phase modulation and stored frequency components of the signal as well as between the spatial-phase modulation and the frequency distribution of the stored spin-wave due to the imposed AC-Stark shift.

We thank the Reviewer for this very elaborate yet precise commentary. We are pleased to hear that our work brings novel ideas and the progress reported is substantial. We are also grateful for identifying our benchmark tools and analyses a good reference and guide for future works in the topic. The criticism of the structure and content of the manuscript has motivated us to vastly modify and extend the article with more intuitive explanations, accessible to a broader audience.

In the revised manuscript we now include an additional section that introduces the basics of temporal imaging in GEM. The section starts with a simple explanation of the GEM protocol along with relevant figure (Fig. 3(a)) showing evolution of the atomic coherence in the real (z) and wavevector (kz) coordinates for exemplary input pulses. We also highlight there the correspondence between the spatial and temporal phase terms of light and atomic coherence by directly comparing the shape of optical and atomic coherence on a common plots. Next, we introduce the crucial components needed for temporal imaging and we provide information on how each part is implemented in the GEM. For these we also show an exemplary TI setup performing a Fourier transform of the input pulse and we draw it's real-space imaging analog (Fig. 3(b)). Moreover, the Wigner function formalism is now first introduced for the much simpler TI setup and each transformation is extensively explained with the help of the annotated phase-space map. The same applies to the Figure 4 (previously Figure 2), that has been updated to include annotations, color scales, and new schematic representation of the protocol.

Similarly, decomposing an input waveform into symmetric and anti-symmetric components is the key idea of the PuDTAI interferometer, which is already mentioned in the beginning of the paper, but the reason and intuition behind this principle is not obvious to a reader who is non-specialist in super-resolution spectroscopy. Actually, just in the middle of the paper, the authors describe this fact clearly with a simple explanation and basic mathematical expressions (page-4, Eqns 4,5). In this case, I suggest the authors to introduce this part before the description of the PuDTAI technique, which would put everything in a more transparent context for easy understanding.

We merged the introductory part of the Protocol section into the Information in superresolution section that is now before the Protocol section. We hope that the current arrangement of the two parts is more natural and easier to follow.

As mentioned above, the presented analyses of the experimental results are very systematic and informative. On the other hand, the authors could use a bit more “introductory language” when mentioning the basic concepts, including the Rayleigh curse, Cramer-Rao bound, and Fisher Information. In addition, the calculated/measured basic statistical parameters (e.g., max. likelihood estimator, variance) should be defined and recapped in such way that the results presented in 3a-c should give a clear insight into what is achieved in this study. For example, according to Fig. 3a, the raw estimates of the spectral separation parameter (ϵ) yield the same value for QMTI and PuDTAI, but with vastly different uncertainty. The implication of this result needs a bit more clarification in practical terms (e.g., how this parameter indicates the superiority of PuDTAI over QMTI in a certain measurement setting).

In the revised manuscript we moved the introductory section “Information in spectral resolution” to the beginning of the manuscript. The section introduces the basic concepts used throughout the rest of the work and sets the stage of the problem. Moreover, we extended the estimation theory part to include more intuitive explanations for the Fisher information and Cramer-Rao bound. We hope that with those modifications the section is more reader-friendly and provides insight into the topic for non-specialist.

The variance of the estimator is estimated using a bootstrapping technique on 1.5×10^5 collected counts for each separation value ϵ . As we describe in the manuscript, for each setting the estimator is evaluated 10^3 times on 1.5×10^5 randomly chosen counts (with repetitions). The variance is then calculated from those 10^3 estimation results. The same procedure is used in the case of direct imaging technique. To emphasise the difference of the precision of the two protocols and make it visible on the raw estimates plot in the Figure 5a (previously Fig. 3a) we normalized the obtained uncertainties to 10 collected photons, while the raw estimates are shown for the full statistics. This information is included in the caption of the figure as well as in the main text.

We reformulated the part describing the procedure to be more clear and exact. Moreover, we now provide implication of the obtained results just after they appear in the manuscript. We have chosen to build the precision comparison of the two techniques (DI and PuDTAI) on the variances, as they scale linearly with the number of photons used. The superresolution parameter introduced in the final part of the manuscript is just the extension of such comparison that sets the analogous yet ultimate improvement based on Fisher information. Additionally, to clarify things even more, in the Methods section we now provide exact procedures for the maximum likelihood estimators used in the article.

In short, I think that this study by M. Mazelanik et al meets the essential publication-criteria of Nature Communications from the aspects of scientific quality, technical advance and potential impact. However, I believe that the manuscript should be more accessible to a general audience, considering the multi-disciplinary nature of this journal. In this regard, following a potential revision that addresses the above points, I would be happy to recommend this article for publication in Nature Communications.

We are grateful for hearing that our work meets the essential publication-criteria of the prestigious *Nature Communications* journal. We hope that the new version of the manuscript is more accessible to a general audience and fits into the multidisciplinary paradigm of the journal.

In the revised version, I also would like the authors to address the following minor comments and points:

1. In the introduction, the authors mention that long coherence-time of spin-wave memory is an important factor to extract the relevant spectral information from light in an optimal way. I would like the authors elaborate on this fact (for example, in discussion section), expressing how the longer storage times would boost the performance of the current PuDTAI implementation.

The storage or coherence time essentially limits the achievable size of the temporal aperture that sets the Fourier limit for direct imaging approach. In this case pulses longer than the aperture time are clipped which spoils the output spectra and thus limits the separation estimation precision. This is taken into account in the comparison Figure 6 where each DI spectrometer is represented as a curve that exhibits a roll-off behavior for specific signal bandwidth given approximately by the inverse of the temporal aperture. Similarly, in the case of the PuDTAI protocol, longer coherence times allows mapping of longer signal pulses onto the atomic coherence and thus enables resolving signals of narrower bandwidth. However, for given setting of the signal bandwidth broader than the inverse of the storage time, the storage time has no impact on the PuDTAI relative separation estimation performance.

2. The authors often use the terminology of the space-domain imaging for explaining the frequency domain operations. In this respect, I think that the analogy between a spatial aperture and spectral aperture (controlled by quantum memory) is not clearly described (on page-2, right column, first paragraph). I would suggest authors to make a simple imaging diagram (similar to Fig. 2a), showing to one-to-one correspondence of the relevant terms (e.g imaging, focusing etc.) between the two domains.

As already mentioned above, in the new introductory section we added a figure (Fig. 3b) representing the basic temporal imaging setup. Each component of the setup is annotated there which we hope clarifies the link between temporal and spatial domain.

3. Fig 1a is a really nice illustration for describing time-inversion interferometry, but Fig 1b needs a bit more explanation (perhaps in the caption). Especially, the real function of the GEM coils is not described until Method section. Also, it took a while for me to interpret what the blue and red curves flying through the coils. So, they should be specified. In addition, there should be a color-scale for the plots shown in Fig. 2b

We have updated Figure 1 to be more self-explanatory and divided it into two separate figures (Fig. 1 and Fig. 2). In Particular, we added a simplified experimental sequences that shows the use of GEM coils and explicitly links the optical field shapes flying through the coils with input and output signal pulses. Additionally, the function of the GEM coils is now explained in the new section mentioned above that describes the temporal imaging in GEM.

As mentioned in the answer to the first comment, we remade the Figure 2 (now Figure 4) and it now includes all the previously missing parts.

4. Fig 3a seems to make an analogy between the space and time/frequency domain pictures of a time-inversion interferometer. But there is no explanation of what each image-piece and labeling represent and refer to. Especially, it was a bit a puzzle for me to understand the meanings of the colored curves (inside the pulse envelopes), the pair of blue objects in the middle, and labels of the interferometer's inputs (until I saw a similar figure (fig.7) with a bit more explanation in Methods).

We understand that this comment is about Fig. 2a. As mentioned above, we upgraded Fig. 2 (which is now Fig. 4) which is now consistent with the new Fig. 3. The collection of curves representing the set of possible realizations of the signal shape in the spectral and temporal

domain has been replaced with the more convenient decomposition of the signal pulse to the symmetric and antisymmetric components. The figure is now an extension of the Fig. 3b which with the additional legend and annotations provided makes it much easier to understand. We hope that this new approach makes our explanation much more extensive.

5. I think that the timing sequence, illustrated in Fig-5, is quite useful for understanding what is going on with the experimental implementation. The authors may consider moving it (or its simplified version) to the main text.

We remade some parts of the Figure 1, divided it into two separate figures (Fig.1 and Fig. 2) and included more annotations. Especially, as suggested by the Reviewer, we added a simplified version of the experimental sequences that links particular elements from the figure with the implementation of the given protocol (GEM, TI or PuDATI). The caption of each figure has been updated accordingly.

Reviewers' Comments:

Reviewer #2:

Remarks to the Author:

The authors have considerably improved the manuscript. They added a section that explains the GEM and TI methods, improved the figures (figures 1 and 2 are great) and added figure 3, which is also an excellent figure that clarifies the relevant explanations.

The paper is much better now, however it's still very dense and challenging.

I think that adding a bit more details and clarifications can make it more approachable:

1. While figs 1,2,3 are great, I still struggle to understand fig. 4.

The input should be very close to a Gaussian, so the original Wigner function needs to be approximately Gaussian, right?

What is being plotted in the storage is "the two halves of the signal pulse mapped in a symmetric manner on separate Fourier (k_z) components while overlapping in the longitudinal coordinate z ".

Maybe you can show in a picture, or in a formula, what's the original signal (Wigner function) and how is it transformed to this Wigner function.

It's also hard to see the difference between the left Wigner function and the right one. Then in the two AC-S stages it seems

like the arrows act differently on the symmetric and anti-symmetric parts, what's the reason for this? Which equations describe this?

2. The presentation and explanation of the GEM, TI are good, but due to the lack of any equations I'm left with some questions:

a. Storing the field in the coherence terms: Is there any reference that shows the derivation of this? 3 levels are

necessary here? it cannot work with 2-levels?

b. "To restore the signal light, the gradient is switched to the opposite $g \rightarrow -g$ and the control field pulse is applied again." (p. 4) Maybe it's a basic fact in quantum optics, but why is this correct? A reference with a derivation would suffice.

Minor comments:

1. Captions fig. 4: "and makes the two half" needs to be "and makes the two halves"

2. Captions fig. 2: "The same setup is used the implement a the simple GEM".

Reviewer #3:

Remarks to the Author:

The authors have addressed all of my previous points satisfactorily and improved the manuscript significantly with several important changes. I think that the revised version is now well structured and easier to follow. The new figures and additional explanations make it much more accessible to non-experts in the field. I am happy to recommend it for publication in Nature Communications.

Towards publication I have some minor suggestions that the authors may consider to implement.

- In Figure 3a, the y-axis scale for k (wavevector) on the right-hand side is not aligned (i.e, slightly shifted) with respect to the left-hand side. Therefore, the amplitude profile of k (right panel) doesn't fully match the color scale of the evolving k vectors, which may lead to misinterpretation. I would suggest the author to remove the offset between the left and right k -axes.

- In order to make a better connection between the descriptions of the PuDTAI protocol (represented in the k - z domain) and the temporal imaging scheme, the authors may consider to

show the Wigner representation both in the k - z and w - t domains in Figure 3d (instead of showing it only in the w - t domain).

- The second paragraph of Results Section (A more conventional, but equally relevant...) ends of a brief discussion on the role of control field in setting the aperture. I think that it is a bit too early to mention about this, as all of the relevant concepts are defined and introduced in the following subsections of the paper. I will suggest the authors either to simplify these wordings or to omit this discussion at this place.

- I will recommend a slight-change in the title : " Optical-domain spectral super-resolution via a quantum memory based time-frequency processor"

Reviewer #2:

The authors have considerably improved the manuscript. They added a section that explains the GEM and TI methods, improved the figures (figures 1 and 2 are great) and added figure 3, which is also an excellent figure that clarifies the relevant explanations.

The paper is much better now, however it's still very dense and challenging.

I think that adding a bit more details and clarifications can make it more approachable:

We thank the Reviewer for acknowledging our work and appreciating the improvements and new figures that we made during the last revision. Below we address the remaining comments that helped us to improve the manuscript even more.

1. While figs 1,2,3 are great, I still struggle to understand fig. 4.

The input should be very close to a Gaussian, so the original Wigner function needs to be approximately Gaussian, right?

That is true for the symmetric (gaussian) case, the Wigner function for the antisymmetric shape (Hermite-Gaussian) has a different shape. To clarify on that we added the signals' (original) Wigner functions before the storage in Fig. 4.

What is being plotted in the storage is "the two halves of the signal pulse mapped in a symmetric manner on separate Fourier (k_z) components while overlapping in the longitudinal coordinate z ".

Maybe you can show in a picture, or in a formula, what's the original signal (Wigner function) and how is it transformed to this Wigner function.

It's also hard to see the difference between the left Wigner function and the right one. Then in the two AC-S stages it seems

like the arrows act differently on the symmetric and anti-symmetric parts, what's the reason for this? Which equations describe this?

The formula for the Wigner function and especially the transformed one is complicated and as we believe would not help in understanding the protocol. Instead of that we now show the input signals' Wigner functions before the storage, and show the transformation arrows in both (symmetric and antisymmetric) cases, which clarifies that the transformation of the Wigner space does not depend on the input signal shape. However, the Wigner distribution after the transformation clearly depends on the input signal shape which is the result of the interference in the Wigner space around $k_z = 0$ coordinate. Moreover in Fig. 4b we now include the temporal aperture that removes the central part of the signal pulse which is especially visible in the symmetric case, and results in non-Gaussian Wigner function after the storage. The changes in Figure 4b are accompanied by a short additional comment in the *PuDATI interferometer* subsection that as we hope resolves the issue.

2. The presentation and explanation of the GEM, TI are good, but due to the lack of any equations I'm left with some questions:

a. Storing the field in the coherence terms: Is there any reference that shows the derivation of this? 3 levels are necessary here? it cannot work with 2-levels?

Indeed the GEM protocol works with 2-levels, however, the three-level system gives additional control i.e. the control field amplitude can be switched only for a certain amount of time to perform partial readout (as we do in our work). The detailed

description of the three-level GEM can be found in the refs [38] (*Optica* **3**, 100107 (2016)), [39] (*Nature* **461**, 241245 (2009)) as well as in the ref. [37] (*Optica* **7**, 203208 (2020)) for our particular setup.

b. "To restore the signal light, the gradient is switched to the opposite $g \rightarrow -g$ and the control field pulse is applied again." (p. 4) Maybe it's a basic fact in quantum optics, but why is this correct? A reference with a derivation would suffice.

This is correct. The magnetic field gradient is used to constantly dephase the coherence during the storage, so multiple light pulses (or time bins) can be stored in the memory. In fact, this is the crucial feature that provides the spectrum-to-position and time-to-wavevector mapping. For the readout of the particular time bin, the associated coherence has to be rephased so contributions to emission from different positions along the ensemble interfere constructively (the phase-matching condition is satisfied). This is achieved by reversing the gradient and letting the coherence rephase during the second half of the storage period. This is in detail described in the references [37-39] in the manuscript (the references that we also link above appear when introducing the GEM protocol).

Minor comments:

1. Captions fig. 4: "and makes the two half" needs to be "and makes the two halves"

Corrected.

2. Captions fig. 2: "The same setup is used the implement a the simple GEM".

Corrected.

Reviewer #3:

The authors have addressed all of my previous points satisfactorily and improved the manuscript significantly with several important changes. I think that the revised version is now well structured and easier to follow. The new figures and additional explanations make it much more accessible to non-experts in the field. I am happy to recommend it for publication in *Nature Communications*.

We are pleased to hear that we addressed all of the Reviewer's concerns and the new version of the manuscript was recommended for publication in *Nature Communications*.

Towards publication I have some minor suggestions that the authors may consider to implement.

- In Figure 3a, the y-axis scale for k (wavevector) on the right-hand side is not aligned (i.e, slightly shifted) with respect to the left-hand side. Therefore, the amplitude profile of k (right panel) doesn't fully match the color scale of the evolving k vectors, which may lead to misinterpretation. I would suggest the author to remove the offset between the left and right k -axes.

The apparent offset was probably a result of the optical illusion caused by the shape of the atomic coherence evolving in time. In the updated version of Fig. 3a. we added a horizontal grid that guides the eyes and as we believe dispels the illusion.

- In order to make a better connection between the descriptions of the PuDTAI protocol (represented in the k-z domain) and the temporal imaging scheme, the authors may consider to show the Wigner representation both in the k-z and w-t domains in Figure 3d (instead of showing it only in the w-t domain).

We updated the Wigner representation to include both, (k_z, z) and (t, ω) coordinates.

- The second paragraph of Results Section (A more conventional, but equally relevant...) ends of a brief discussion on the role of control field in setting the aperture. I think that it is a bit too early to mention about this, as all of the relevant concepts are defined and introduced in the following sub-sections of the paper. I will suggest the authors either to simplify these wordings or to omit this discussion at this place.

We removed the discussion on the role of the control field in setting the aperture from this paragraph.

- I will recommend a slight-change in the title : " Optical-domain spectral super-resolution via a quantum memory based time-frequency processor"

We changed the title to *Optical-domain spectral super-resolution via a quantum-memory-based time-frequency processor.*